# Hard-Negative Prototype-Based Regularization for Few-Shot Class-Incremental Learning

**Seongbeom Park[1], Hyunju Yun[1], Daewon Chae[2], Sungyoon Kim[3], Suhong Moon[3], Minwoo Kang[3], Seunghyun Park[4†], Jinkyu Kim[1†]**

[1]*Korea University*  [2]*University of Michigan, Ann Arbor*  [3]*University of California, Berkeley*  [4]*Soongsil University*
[†]*Corresponding Authors. Email:* `sh.park@ssu.ac.kr` *and* `jinkyukim@korea.ac.kr`

Reviewed on OpenReview: *https://openreview.net/forum?id=xKn7OOvDaR*

## Abstract

Few-shot class-incremental learning (FSCIL)—involving abundant base training data followed by novel classes with limited labeled samples—poses challenges such as catastrophic forgetting and overfitting, leading to significant performance degradation across incremental sessions. As a remedy, recent work focuses on minimizing the interference of embeddings between base and incremental classes. However, previous studies have not explicitly considered variation in discriminative difficulty across samples and classes, leaving room for improvement: we observe that hard-negative (i.e., difficult to discriminate from the label) samples and classes significantly affect FSCIL performance, whereas easy ones have little impact. To this end, we propose a hard-negative prototype-based regularization approach that enhances discrimination between similar classes by imposing a penalty margin between each sample and its most similar class prototypes based on cosine similarity. To select hard-negative prototypes, we explore two distinct mining strategies: dynamic selection that leverages the model's decision boundary, and static selection that utilizes a pre-defined class-wise similarity matrix derived from external sources such as pre-trained models. We evaluate our approach on three widely used benchmarks, miniImageNet, CIFAR100, and CUB200, achieving state-of-the-art performance on each. Comprehensive analyses demonstrate that our proposed method enhances intra-class cohesion and inter-class separability of embeddings, both of which are crucial for FSCIL to better accommodate novel classes.

## 1 Introduction

Few-shot class-incremental learning (FSCIL) has recently gained significant attention due to its potential for a variety of real-world applications (Tao et al., 2020). FSCIL training process typically consists of two phases: (1) the base session, in which a model is trained on base classes with a large amount of labeled data, and (2) the incremental session, where the model is fine-tuned on novel classes using few labeled samples. FSCIL poses two challenging tasks that need to be addressed jointly: (i) class-incremental learning, in which a model must learn newly added classes without forgetting previously acquired knowledge (i.e., catastrophic forgetting (McCloskey & Cohen, 1989)); and (ii) few-shot learning, requiring the model learns new knowledge from a small number of training examples while avoiding overfitting to the new classes and generalizing to unseen test samples.

Various methods have been proposed for FSCIL, with a focus on minimizing interference of embeddings between base and incremental classes. In this context, two main approaches have been explored—prototype learning and embedding space learning—either separately or in combination. Prototype learning aims to learn distinguishable and class-representative vectors that can generalize to unseen novel classes, as demonstrated by recent efforts such as human-cognition-inspired prototypes (Yao et al., 2022) and orthogonal pseudo-targets (Ahmed et al., 2024). Additionally, embedding space learning seeks to construct compact intra-class clusters while enforcing large inter-class margins (Peng et al., 2022; Zou et al., 2022; Song et al., 2023), thereby preserving sufficient space for incoming novel classes.

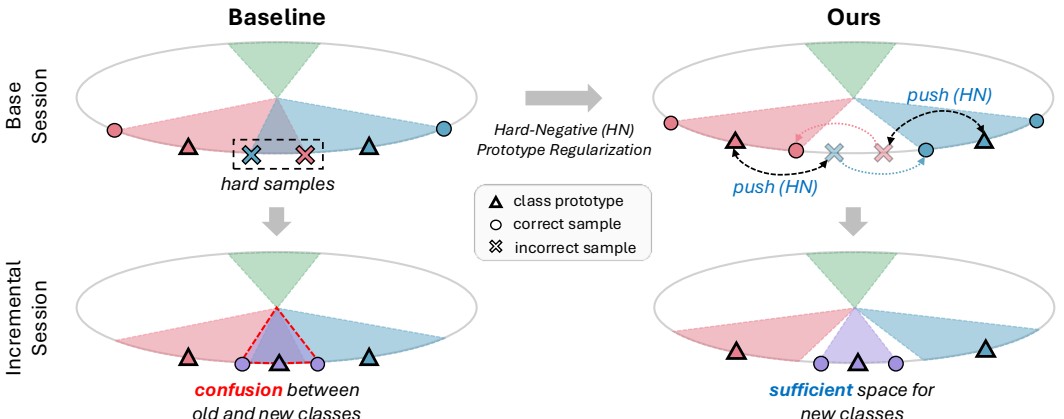

Figure 1: An overview of the proposed hard-negative prototype regularization approach. Previous methods which overlook variation in discriminative hardness across samples and classes often struggle with hard examples, leading to confusion between old (red, green, blue) and new (purple) classes. Our method leverages hard-negative prototypes of each sample to impose a penalty margin, effectively pushing hard samples away from confusing classes and creating sufficient space to accommodate novel classes in incremental sessions.

Nonetheless, while prior work has effectively mitigated catastrophic forgetting in FSCIL, they often overlook the variation in discriminative difficulty across samples and classes, e.g., visually similar classes or ambiguous examples near decision boundaries, leaving room for improvement. As illustrated in Figure 1, a key factor shaping a model's decision boundary is the presence of *hard samples*, which reside far away from their respective class centers (i.e., prototypes in our context). These hard samples negatively impact FSCIL in two ways: (1) they cause confusion in the model's predictions, resulting in overall performance degradation, and (2) they occupy regions in the embedding space, potentially hindering the adaptability of novel classes and causing the forgetting of previously learned classes.

To support our motivation, we analyze the variation in sample and class difficulty and observe that hard-negative samples and classes are key to success in FSCIL. Specifically, after training on the base classes, we classify their test samples into hard and easy groups by selecting the top-$k$ samples that are farthest from and closest to their prototypes, respectively. As shown in Figure 2, easy samples not only achieve superior performance but also experience little or no catastrophic forgetting (compare dark vs. light for each color of easy samples). In contrast, hard samples exhibit poor performance, with significant forgetting occurring after learning incremental classes. These tendencies indicate that hard samples are the main cause of confusion and forgetting in FSCIL. To further analyze this confusion, we evaluate the top-$k$ accuracy of models and compute the delta—the difference between accuracies for adjacent $k$ values—as illustrated in Figure 3. The delta graphs show an exponentially decreasing trend, implying that most confusions are concentrated among a small number of classes. We define these classes as hard-negative classes and leverage their prototypes to regularize samples, thereby minimizing interference among embeddings of different classes. A detailed analysis of the difficulty variation is provided in Section 3.

Inspired by these observations, we propose a novel hard-negative prototype regularization (HNPR) approach by identifying the top-$k$ most similar class prototypes for each sample and impose a penalty margin between them. By applying this penalty margin, the model is trained to focus on distinguishing similar classes, thereby pushing them apart and yielding a larger margin in the embedding space for novel classes. To select these hard-negative prototypes, we explore two distinct mining strategies based on the scope of similarity comparison. Dynamic selection leverages the model's decision boundary during training, while static selection utilizes a pre-defined class-wise similarity matrix derived from external sources such as pre-trained visual or textual models. We validate the effectiveness of our approach against three widely used benchmarks— miniImageNet (Vinyals et al., 2016), CIFAR100 (Krizhevsky et al., 2009), and CUB200 (Wah et al., 2011)— achieving state-of-the-art performance across all benchmarks. Extensive analysis demonstrates the intra-class cohesion and inter-class separability of the learned embedding space, which supports the superior performance. Our contributions are summarized as follows:

- We analyze the variation in difficulty among samples and classes, demonstrating that the poor performance and significant forgetting exhibited by hard samples and classes pose critical challenges for effective FSCIL.

- We propose a novel hard-negative prototype regularization approach that imposes a penalty margin on similar classes, thereby guiding the model to focus on their distinctions. Additionally, we explore two mining strategies—dynamic selection and static selection—for identifying hard-negative prototypes based on the scope of similarity comparison.

- We conduct extensive experiments on three widely used benchmarks, demonstrating the superiority of our approach and achieving state-of-the-art performance across all evaluated benchmarks.

## 2    Related Work

**Few-Shot Class-Incremental Learning (FSCIL).** FSCIL presents a challenging problem that involves incrementally adapting to a stream of data with few labeled samples (Tao et al., 2020). Various approaches have been proposed and can be classified as follows: (i) meta-learning-based methods (Chi et al., 2022; Zhu et al., 2021), (ii) prototype-based methods (Hersche et al., 2022; Zhu et al., 2021; Zhuang et al., 2023), (iii) dynamic network-based methods (Tao et al., 2020; Kang et al., 2023; Yang et al., 2022), (iv) replay-based methods (Liu et al., 2022), and (v) embedding space-based methods (Kim et al., 2023; Cheraghian et al., 2021; Song et al., 2023; Zhou et al., 2022a; Oh et al., 2024). Among these, for embedding space-based methods—which have recently shown remarkable success—aim to achieve intra-class compactness and inter-class discriminability to minimize feature interference between old and new classes. In this context, ALICE (Peng et al., 2022) utilized an angular margin loss, complemented by augmentation and balanced testing, to obtain well-clustered features. Furthermore, SAVC (Song et al., 2023) applies predefined transformations to generate virtual classes and uses supervised contrastive learning in a multi-semantic fantasy space to improve separation between base and future novel classes. Although these studies validate the effectiveness of a discriminative feature space, variations in data difficulty have not been explicitly considered. Therefore, we propose a novel method that encourages the model to better distinguish between similar classes by leveraging an additional penalty margin for hard-negative prototypes.

**Metric Learning with Hard-Negatives.** Unlike traditional learning methods that directly predict labels or values, metric learning focuses on capturing the relationships within a data distribution through a similarity-aware distance function. Earlier work, such as contrastive loss (Chopra et al., 2005) and triplet loss (Wang et al., 2014; Hoffer & Ailon, 2015), utilizes margins to enhance model learning ability by ensuring that negative pairs are sufficiently separated in the embedding space. Additionally, angular distance metrics, employed in large-margin softmax loss (L-Softmax) (Liu et al., 2016) and its improvements, further refine feature discrimination through angular regularization (Liu et al., 2017) and additive margins (Wang et al., 2018; Deng et al., 2019). Hard-negative mining plays a crucial role in metric learning by targeting the most challenging negative examples—those closest to the anchor in the embedding space. The use of (semi-)hard-negative mining has demonstrated significant performance improvements across various domains, including face recognition (Schroff et al., 2015), object detection (Shrivastava et al., 2016), image classification (Suh et al., 2019), and graph contrastive learning (Xia et al., 2021). Furthermore, similar metric- and hard-example mining-based approaches have been extensively explored in the few-shot learning literature (Mandalika, 2025; Roy et al., 2022; Li et al., 2021), where prototype- and relation-based methods leverage embedding distances to generalize from only a few examples. While previous works have primarily focused on sample-wise mining within mini-batches (Xuan et al., 2020; Robinson et al., 2020), we extend this approach by exploring class-wise hard-negative mining based on representative prototypes, which are commonly employed in continual learning scenarios (Hersche et al., 2022; Zhu et al., 2021). This prototype-based strategy closely aligns with class-level proxy mining principles (Movshovitz-Attias et al., 2017; Kim et al., 2020), offering a theoretical foundation. To the best of our knowledge, this is the first approach to leverage the hard-negative concept within the FSCIL literature.

## 3    Hard-Negatives in FSCIL

In this section, we demonstrate that hard-negative samples and classes are crucial for effective FSCIL, motivating our hard-negative prototype regularization approach.

### 3.1 Preliminaries

**Problem Formulation.** We follow the standard FSCIL experiment setup, which consists of two main phases: (i) a base session and (ii) multiple incremental sessions. In the base session, a model is first trained with sufficient training data. Then a limited number of training data (thus called few-shot) is provided in the incremental sessions. Note that, in the incremental sessions, data from previous sessions are inaccessible; this presents the challenges of retaining the knowledge learned from earlier sessions. Formally, we assume $m$-step FSCIL tasks where we have $m + 1$ training data splits for the base and incremental sessions, i.e., $\mathcal{D} = \{\mathcal{D}^{(0)}, \mathcal{D}^{(1)}, \ldots, \mathcal{D}^{(m)}\}$. In session $i \in \{0, 1, ..., m\}$, the goal is to learn a function $f : \mathcal{X} \to \mathcal{Y}$, given the training data $(x, y) \in \mathcal{D}^{(i)}$, where $x \in \mathcal{X}^{(i)}$ is an input sample and $y \in \mathcal{Y}^{(i)}$ is its corresponding label. Note that the label spaces for different sessions are disjoint, i.e., $\mathcal{Y}^{(i)} \cap \mathcal{Y}^{(j)} = \emptyset$ for $i \neq j$. Thus, a model should learn new classes in each incremental session while retaining knowledge of old classes without access to past samples. An incremental session is often organized in a $C$-way $K$-shot setting, where each session has training samples from $C$ classes, with $K$ samples per class, providing a total of $C \times K$ training samples per session. Concretely, at the $i$-th incremental session, we optimize the following learning objective:

$$\text{minimize} \quad \mathbb{E}_{(x,y) \sim \mathcal{D}^{(0)} \cup \mathcal{D}^{(1)} \cup \cdots \cup \mathcal{D}^{(i)}} [\mathcal{L}(f(x; \mathcal{D}^{(i)}), y)], \tag{1}$$

where we minimize the empirical risk over all the seen classes up to session $i$.

**Prototype-Based Classification.** Prototype-based classification is a widely used approach in FSCIL literature due to its simplicity and effectiveness in handling limited data for each class (Zhu et al., 2021; Hersche et al., 2022; Zhuang et al., 2023). Following previous work (Peng et al., 2022), we define the prototype $P_c$ of class $c$ as the average feature of training samples. Classification is performed by calculating the cosine similarity between an input feature and the prototypes, which can be formalized as follows:

$$\hat{y} = \arg\max_c \frac{f(x) \cdot P_c}{\|f(x)\| \|P_c\|}, \tag{2}$$

where $\hat{y}$ denotes the predicted class, $\cdot$ represents the dot product, and $\|\cdot\|$ is the Euclidean norm. During the base session training, instead of recalculating prototypes at every epoch, we introduce randomly initialized, learnable weights $W$, which act as pseudo-prototypes in place of $P$. These pseudo-prototypes are updated dynamically through backpropagation, allowing them to adapt continuously to the classification task. The training objective is to minimize the loss by optimizing these learnable pseudo-prototypes. After the base session training, we compute the prototypes for both the base and incremental session classes using the frozen backbone.

**Baseline.** We set a baseline approach using angular margin loss, which aims to reduce the overlap between features from different classes. Previous studies (Liu et al., 2017; Wang et al., 2018; Deng et al., 2019) have proposed various methods to enforce a margin on the angular decision boundary, commonly referred as angular margin loss $\mathcal{L}_{\text{AM}}$. Variants of the angular margin loss function can be formulated based on choices of (i) the scale factor $s$ and (ii) margins $m_1$, $m_2$, and $m_3$ that define the angular decision boundary:

$$\mathcal{L}_{\text{AM}} = -\frac{1}{N} \sum_{i=1}^{N} \log \frac{e^{s(\cos(m_1 \theta_{y_i} + m_2) - m_3)}}{e^{s(\cos(m_1 \theta_{y_i} + m_2) - m_3)} + \sum_{j \neq y_i} e^{s(\cos(\theta_j))}}, \tag{3}$$

where $\theta$ denotes the angle between the input feature and the class prototype, $N$ is the number of samples, and $y_i$ represents the ground-truth class. Note that the additive margin $m_3$ applied to the cosine values is typically used in the FSCIL task. This is because (i) the multiplicative margin $m_1$ often leads to unstable training issues by making the target logit curve steeper (Deng et al., 2019) and (ii) the additive margins $m_2$ and $m_3$ are numerically similar. The angular margin loss for the FSCIL task (Peng et al., 2022) can be formulated as follows:

$$\mathcal{L}_{\text{AM}} = -\frac{1}{N} \sum_{i=1}^{N} \log \frac{e^{s(\cos(\theta_{y_i}) - m)}}{e^{s(\cos(\theta_{y_i}) - m)} + \sum_{j \neq y_i} e^{s(\cos(\theta_j))}}. \tag{4}$$

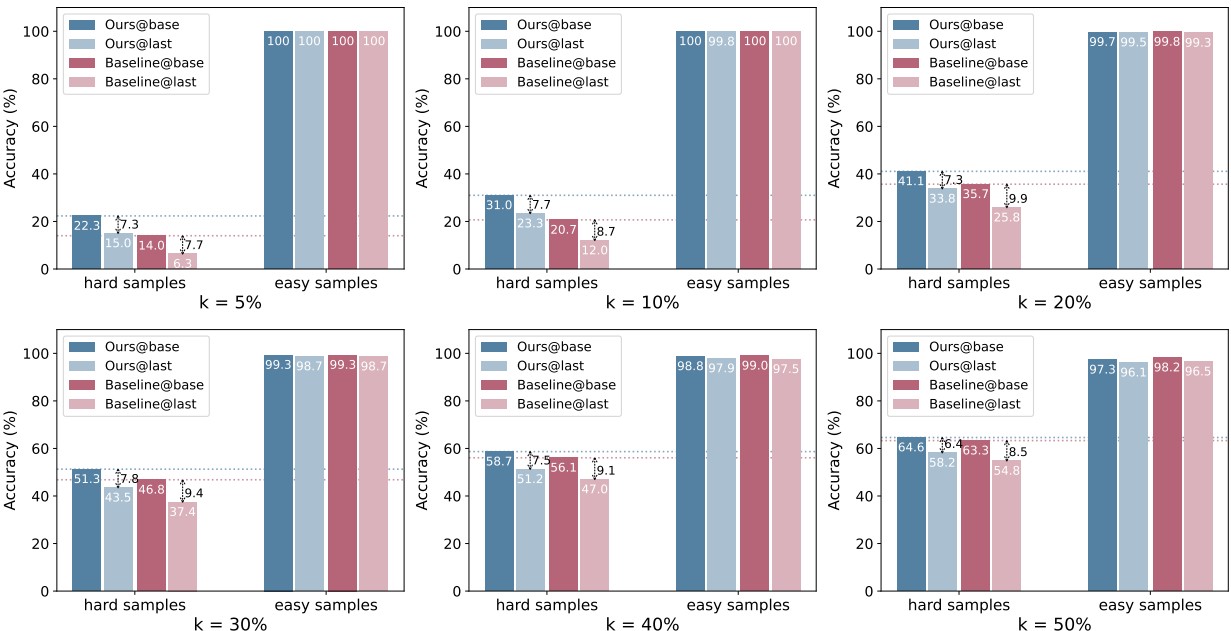

Figure 2: An analysis on the impact of difficulty variation of samples in FSCIL. After training base classes, we divide the test samples of these classes into two groups using the cosine distance from their corresponding class prototypes: hard samples (with the top-$k$ largest distances) and easy samples (with the top-$k$ smallest distances). The baseline is trained using angular margin loss (Peng et al., 2022), without hard-negative prototype regularization. We report the accuracy after the base session (dark colors) and the last session (light colors) for each method. Hard samples are observed as the main cause of confusion and forgetting, whereas easy samples have little effect on overall performance.

## 3.2 Impact of Hard-Negatives in FSCIL

**Easy Samples vs. Hard Samples.** By analyzing variations in sample difficulty, we observe that hard samples are key factors in both performance and forgetting in FSCIL. Specifically, after training on the base classes, we divide the test samples of the base classes into two groups: *hard samples*, defined as the top-$k$ samples with the largest distances from their corresponding prototypes, and *easy samples*, defined as the top-$k$ samples with the smallest distances. We then report the accuracy for each group after the base session (i.e., right after learning the classes) and after the last session (i.e., after learning all incremental classes). As shown in Figure 2, easy samples show superior performance and exhibit little or no catastrophic forgetting, whereas hard samples not only perform poorly but also suffer from significant forgetting during incremental sessions. These trends become even more pronounced when using smaller values of $k$, further underscoring that *hard samples* are critical to the success of FSCIL. Note that our proposed method reduces the forgetting of hard samples between the base and last sessions, which will be discussed in the following sections.

**Confusion in Hard Classes.** Given that hard samples are the main source of confusion in FSCIL, we further analyze class-level confusions and show that misclassifications predominantly occur within a small number of challenging classes. To elaborate, we evaluate the accuracy with top-$k$ predictions (i.e., Acc@$k$) and compute the difference between accuracies for adjacent $k$ values (i.e., $\Delta = \text{Acc@}k - \text{Acc@}k{-}1$, $k > 1$). The resulting delta value reflects the incremental improvement in accuracy when adding one more prediction, and a larger delta indicates that the model exhibits greater confusion among the challenging classes. As shown in Figure 3, in both base and last sessions, the overall trend of delta graphs is exponentially decreasing—implying that most of the confusions are concentrated among small number of challenging classes. Note that the delta value does not designate a fixed set of challenging classes: e.g., some sample in class A may be confused with class B, while others in class A may be confused with class C. These varied confusions

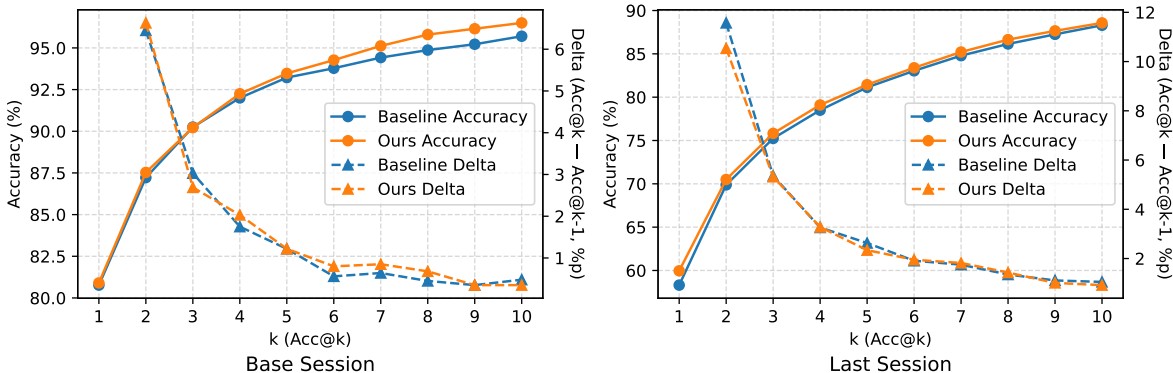

Figure 3: An analysis on the confusion among classes in the base and last sessions using top-$k$ accuracy (i.e., accuracy with top-$k$ predictions, denoted as Acc@$k$) and delta (i.e., the difference between accuracies for adjacent $k$ values for $k > 1$). The same models as in Figure 2 are used for this analysis. The delta graphs (triangle markers) exhibit an exponentially decreasing trend, indicating that most confusions are concentrated among a small number of classes.

are aggregated into a single delta value, reflecting the overall uncertainty within the class. Therefore, we define these challenging classes as hard-negative classes on a per-sample basis and leverage their prototypes to regularize each sample.

# 4 Hard-Negative Prototype Regularization

We propose a novel approach that emphasizes the distinctions between hard-negative classes while incorporating schemes for hard-negative prototype mining. In our baseline, the model is expected to learn two key aspects to minimize the loss function due to the margin effect: (i) the angle between positive pairs should be minimized to reduce intra-class variance (attractive force, $\theta_{y_i}$), and (ii) the angle between negative pairs should be maximized to improve inter-class separation (repulsive force, $\theta_j$). A major limitation, however, is that the repulsive force is uniformly exerted on all negative pairs, meaning the margin $m$ is applied equally regardless of the pairs. Considering the varying degrees of similarity among classes, this approach may restrict the model's ability to distinguish between similar classes, potentially leading to sub-optimal performance. This limitation is further confirmed by our observation in Section 3.2, where most confusions are concentrated among the top-$k$ challenging classes. To address this issue, we identify similar classes and leverage their prototypes to regularize the learning of challenging classes by imposing an additional margin.

## 4.1 Hard-Negative Prototype Mining

To enable a model to better distinguish challenging classes, we introduce two hard-negative mining approaches based on the scope of similarity comparison: dynamic selection and static selection. Our mining process relies on the cosine similarity between the input feature and class prototypes. Dynamic selection leverages the model's decision boundary during learning, whereas static selection utilizes a pre-defined classwise similarity matrix from external sources (e.g., pre-trained models). Figure 4 illustrates the overall flow of these different mining strategies.

**Dynamic Selection.** Assuming that the model is optimized via stochastic gradient descent, the similarity distribution of an input sample varies with each gradient step. Therefore, selecting similar classes through batch-wise similarity calculation can reflect the decision boundary at the current optimization step. Given a set of training batches $\mathcal{B}$ and the set of classes $\mathcal{C}$, the similarity score $S_{i,c}^{(b)}$ between a sample $x_i$ in a mini-batch $b \in \mathcal{B}$ and the pseudo-prototype $W_c \in \mathbb{R}^d$ of class $c$ can be computed as follows:

$$S_{i,c}^{(b)} = \frac{f(x_i^{(b)}) W_c^\top}{\|f(x_i^{(b)})\| \|W_c\|^\top}, \tag{5}$$

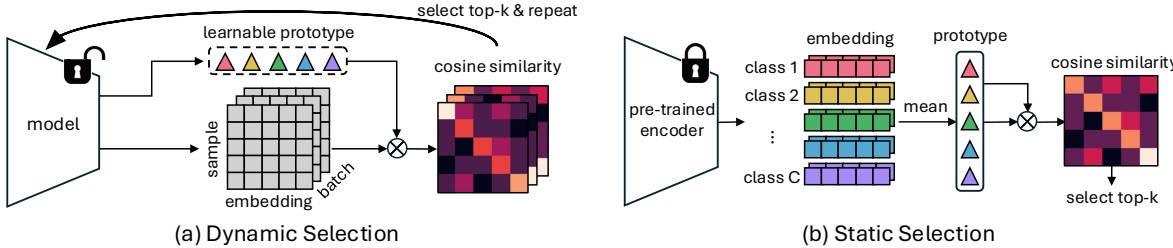

(a) Dynamic Selection        (b) Static Selection

Figure 4: Overview of proposed hard-negative class mining strategies: (a) dynamic selection, and (b) static selection. The dynamic selection approach repeatedly selects the top-$k$ similar classes in a batch-wise manner, considering the continuous shift of similarity distribution at each gradient step. In contrast, the static selection approach leverages a pre-trained encoder to compute similarities, assuming a fixed similarity distribution for selecting hard-negative classes.

where $f(x_i^{(b)}) \in \mathbb{R}^d$ denotes the feature vector. Finally, the top-$k$ similar classes of an input sample are dynamically defined in a batch-wise manner as follows:

$$\mathcal{C}_{(i,k)}^{\text{sim}} = \{c \in \mathcal{C} \mid c \in \text{TopK}(S_i^{(b)}, k)\}, \tag{6}$$

where $S_i^{(b)} \in \mathbb{R}^{|\mathcal{C}|}$ denotes the similarity vector of $x_i$, and the TopK function returns the classes with the $k$ largest similarity scores, excluding the true class $y_i$. Note that under this strategy, similar classes for an input sample may vary across optimization steps because the learnable pseudo-prototypes $W$ are periodically updated. Furthermore, the similar classes for two inputs from the same class can differ, i.e., $\mathcal{C}_{(i,k)}^{\text{sim}} \neq \mathcal{C}_{(j,k)}^{\text{sim}}$ for $i \neq j$ and $y_i = y_j$, since the similarity is measured at the instance level. We empirically observe that this variation helps improve the model's generalizability, as discussed in detail in the Section 5.

**Static Selection.** Apart from dynamic selection, which is based on the current decision boundary, selecting similar classes by leveraging the semantics of a pre-trained encoder on large-scale data could be another option. In this work, we explore two static selection strategies based on the data modality: (i) using a pre-trained visual encoder, and (ii) employing a pre-trained textual encoder. To quantify similarities between classes, we define class-representative prototypes (denoted as $P_c$ for class $c$) using the outputs from each encoder. For the visual encoder, $P_c$ is defined as the average feature of all samples belonging to class $c$, following the traditional approach described in Section 3.1. For the textual encoder, the class prototype is computed using the natural language label of the class. Once the class prototypes are obtained from either modality, we compute the class similarity matrix $S$ using cosine similarity, $S = \bar{P}^\top \bar{P}$, where $\bar{P}$ represents the $L_2$-normalized prototypes. The similarity vector for a sample $x_i$ is obtained by indexing $S$ with its ground-truth class $y_i$, i.e., $S_i = S[y_i]$. Finally, the top-$k$ most similar classes are selected as follows:

$$\mathcal{C}_{(i,k)}^{\text{sim}} = \{c \in \mathcal{C} \mid c \in \text{TopK}(S_i, k)\}, \tag{7}$$

where the TopK function is defined as in Equation (6). In the following sections, we use the dynamic selection as our default mining strategy, while also exploring the effect of static selection in Section 5.2.

### 4.2 Hard-Negative Prototype Regularization in Angular Margin Space

Given the top-$k$ similar classes for a sample $x_i$, we assign an extra margin as a penalty to accelerate the separation between these challenging classes. Concretely, we decompose the term in the denominator of Equation (3)—which imposes a repulsive force between the sample and the negative prototypes—into two parts: one for hard-negative prototypes and the other for non-hard-negative prototypes. For the hard-negative prototypes, we introduce a new regularization term as follows:

$$\mathcal{L}_{\text{HNPR-AM}} = -\frac{1}{N} \sum_{i=1}^{N} \log \frac{e^{s(\cos(\theta_{y_i})-m)}}{e^{s(\cos(\theta_{y_i})-m)} + \sum_{c \in \mathcal{C}_{(i,k)}^{\text{sim}}} e^{s(\cos(\theta_c)+\hat{m})} + \sum_{j \notin \mathcal{C}_{(i,k)}^{\text{sim}}, j \neq y_i} e^{s(\cos(\theta_j))}}, \tag{8}$$

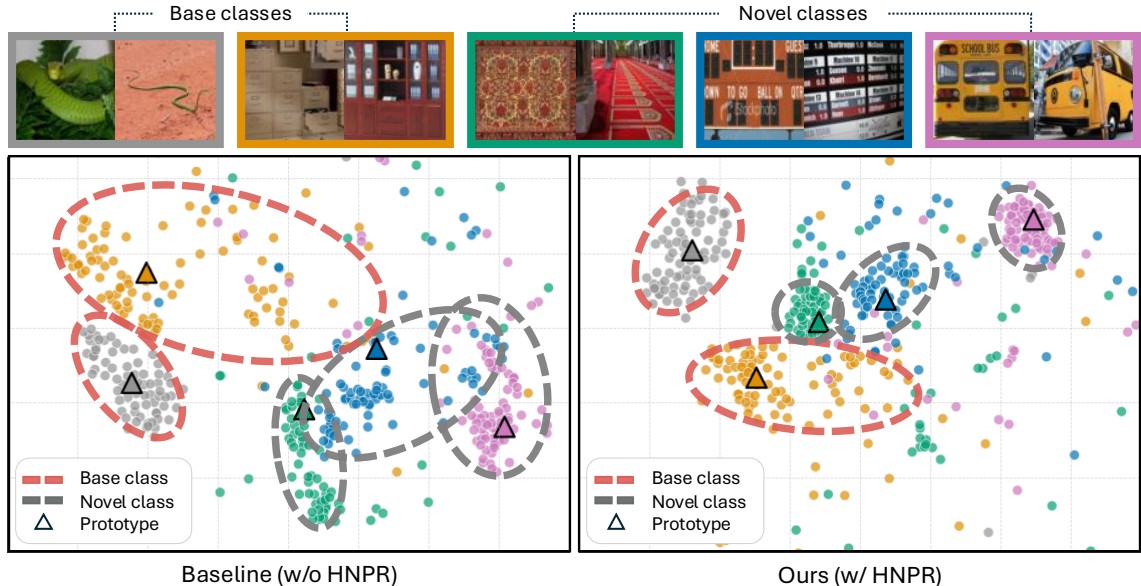

Figure 5: t-SNE (Van der Maaten & Hinton, 2008) visualization comparing the embedding spaces of base and novel classes, alongside the corresponding sample images. Our proposed hard-negative prototype regularization (HNPR) clearly shows better intra-class cohesion and inter-class distinction.

where $\hat{m}$ denotes the extra penalty margin applied to challenging classes. For clarity, we assume that both $m$ and $\hat{m}$ are positive numbers. To minimize the overall loss, a model should deal with not only normal class prototypes ($\theta_j$) but also challenging (hard-negative) class prototypes with a penalty margin (regularized repulsive force, $\theta_c$). Notably, increasing the angular separation between the input feature and similar class prototypes leads the model to capture the nuanced difference between them, contributing to enhancing the model's discriminative capability.

## 5 Experimental Results

**Benchmarks.** We use the following three widely-used FSCIL benchmarks: (1) miniImageNet (Vinyals et al., 2016), (2) CIFAR100 (Krizhevsky et al., 2009), and (3) CUB200 (Wah et al., 2011). miniImageNet and CIFAR100 benchmarks include 100 classes, with each dataset comprising 60K images, while CUB200 benchmark contains over 11K fine-grained images across 200 classes of North American bird species. Following existing works (Kim et al., 2023; Wang et al., 2023; Zhao et al., 2023), we use an 8-step 5-way 5-shot setting for the miniImageNet and CIFAR100 benchmarks, i.e., 60 classes are set as base classes, and the rest is set as new classes used in eight incremental sessions, each of which has five classes and five samples per class. Additionally, we use a configuration of 10 steps with 10 classes and 5 samples per class (10-way 5-shot) for the CUB200 benchmark – in the base session (session 0), half of the total 200 classes are utilized. Then, in the incremental sessions (sessions 1 to 10), the remaining 100 classes are evenly divided into ten separate subsets (with 10 classes per session), with each class being represented by 5 randomly selected examples.

**Implementation Details.** Following the standard setting of prior work (Tao et al., 2020), we use ResNet-18 (He et al., 2016) as the backbone network and train our model from scratch for the miniImageNet and CIFAR100 datasets, while using an ImageNet-pretrained model for the CUB200 dataset. We utilize the augmentation strategies frequently employed in prior FSCIL research (Peng et al., 2022; Yang et al., 2023): (1) resized cropping, (2) horizontal flipping, and (3) color jittering. Our model is trained on a single NVIDIA A100 GPU with an SGD optimizer, using cosine learning rate scheduling with warm-up. Hyperparameters are set with a grid search as follows: $s$ is chosen to be 30 across all datasets. For $m$, we use 0.3 for CIFAR100 and 0.4 for the others. The parameter $k$ (for the top-$k$ selection) is set to 1 for CUB200 and 2 for the others. Lastly, $\hat{m}$ is set to 0.05. Other hyperparameters (e.g., learning rate, weight decay, and batch size) and implementation details are provided in the appendix.

Table 1: Performance comparison with state-of-the-art methods on miniImageNet (Vinyals et al., 2016) benchmark. PD represents the rate of performance drop, calculated as the difference between the accuracies of the base (T0) and last (T8) sessions. All values are presented as percentages (%). We mark the best value in **bold**, and the second best with underline.

| Method | T0 | T1 | T2 | T3 | T4 | T5 | T6 | T7 | T8 | PD($\downarrow$) |
|---|---|---|---|---|---|---|---|---|---|---|
| TOPIC (Tao et al., 2020) | 61.31 | 50.09 | 45.17 | 41.16 | 37.48 | 35.52 | 32.19 | 29.46 | 24.42 | 36.89 |
| CLOM (Zou et al., 2022) | 73.08 | 68.09 | 64.16 | 60.41 | 57.41 | 54.29 | 51.54 | 49.37 | 48.00 | 25.08 |
| ALICE (Peng et al., 2022) | 80.60 | 70.60 | 67.40 | 64.50 | 62.50 | 60.00 | 57.80 | 56.80 | 55.70 | 24.90 |
| FACT (Zhou et al., 2022a) | 72.56 | 69.63 | 66.20 | 62.77 | 60.60 | 57.33 | 54.34 | 52.16 | 50.49 | 22.07 |
| LIMIT (Zhou et al., 2022b) | 72.32 | 68.47 | 64.30 | 60.78 | 57.95 | 55.07 | 52.70 | 50.72 | 49.19 | 23.13 |
| GKEAL (Zhuang et al., 2023) | 73.59 | 68.90 | 65.33 | 62.29 | 59.39 | 56.70 | 54.20 | 52.59 | 51.31 | 22.28 |
| WaRP (Kim et al., 2023) | 72.99 | 68.10 | 64.31 | 61.30 | 58.64 | 56.08 | 53.40 | 51.72 | 50.65 | 22.34 |
| SoftNet (Kang et al., 2023) | 76.63 | 70.13 | 65.92 | 62.52 | 59.49 | 56.56 | 53.71 | 51.72 | 50.48 | 26.15 |
| SAVC (Song et al., 2023) | 81.12 | 76.14 | 72.43 | 68.92 | 66.48 | 62.95 | 59.92 | 58.39 | 57.11 | 24.01 |
| CaBD (Zhao et al., 2023) | 74.65 | 69.89 | 65.44 | 61.76 | 59.49 | 56.11 | 53.28 | 51.74 | 50.49 | 24.16 |
| TEEN (Wang et al., 2023) | 73.53 | 70.55 | 66.37 | 63.23 | 60.53 | 57.95 | 55.24 | 53.44 | 52.08 | 21.45 |
| OrCo (Ahmed et al., 2024) | 83.30 | 75.32 | 71.53 | 68.16 | 65.63 | 63.12 | 60.20 | 58.82 | 58.08 | 25.22 |
| Comp-FSCIL (Zou et al., 2024) | 82.78 | 77.82 | 73.70 | 70.57 | 68.26 | 65.11 | 62.19 | 60.12 | 59.00 | 23.78 |
| Bag of Tricks (Roy et al., 2024) | 84.30 | 79.59 | 75.49 | 71.40 | 68.45 | 65.13 | 62.20 | 60.52 | 59.11 | 25.19 |
| CLOSER (Oh et al., 2024) | 76.02 | 71.61 | 67.99 | 64.69 | 61.70 | 58.94 | 56.23 | 54.52 | 53.33 | 22.69 |
| Ours | 80.90 | 76.06 | 72.24 | 69.92 | 67.27 | 64.96 | 62.07 | 60.91 | 59.96 | **20.94** |

## 5.1 Effect of Hard-Negative Prototype Regularization

**Comparison with State-of-the-Art Approaches.** We compare our method with existing state-of-the-art methods on three FSCIL benchmarks: miniImageNet, CIFAR100, and CUB200. Following previous work (Zhang et al., 2021; Peng et al., 2022; Oh et al., 2024), we employ the performance dropping rate (PD) as the evaluation metric, which represents the difference in accuracy between the base session and the final session. As shown in Tables 1, 3 and 4 (the extended version of Tables 3 and 4 are in the Appendix), our model outperforms other approaches across all benchmarks, achieving new state-of-the-art performance. Furthermore, to assess generalization to modern transformer-based architectures, we compare against the state-of-the-art ASP (Attention-aware Self-adaptive Prompt) (Liu et al., 2024) method on the ImageNet-R (Hendrycks et al., 2021) benchmark. As detailed in Appendix A.1, HNPR outperforms both vanilla ASP and its angular-margin variant—even when using a much larger ViT backbone.

**Effect on Embedding Clusters.** Further, as shown in Figure 5, we employ t-SNE (Van der Maaten & Hinton, 2008) visualizations to qualitatively analyze the impact of our proposed approach. We visualize sample embeddings from both base and novel classes along with their corresponding prototypes (i.e., the average of training sample embeddings). Importantly, our method generally guides the learned representations to map positive pairs closer together while (hard) negative pairs further apart. This finding is further confirmed by Table 2, which measures the average cosine distance between samples and three different types of prototypes.

Table 2: Comparison of average cosine distances between samples and three types of prototypes (positive, negative, and hard-negative).

| Method | Average Cosine Distance | | |
|---|---|---|---|
| | Positive($\downarrow$) | Negative($\uparrow$) | Hard-Negative($\uparrow$) |
| Baseline | 0.237±0.074 | 0.490±0.082 | 0.386±0.075 |
| Ours | **0.231±0.074** | **0.501±0.083** | **0.394±0.077** |

**Effect on Hard Samples and Classes.** As shown in Figure 2, our proposed method reduces the forgetting of hard samples between the base and last sessions (with values displayed immediately to the right of the arrow) regardless of $k$. Notably, for $k \in \{5, 10\}$, the last session accuracy achieved by our method (e.g., 15.0% for $k = 5$) even exceeds the base session accuracy obtained by the baseline (e.g., 14.0% for $k = 5$). Furthermore, in the hard class analysis presented in Figure 3, our model generally demonstrates superior performance in both sessions across various $k$ values (see circle markers), suggesting that hard-negative

Table 3: Performance comparison with state-of-the-art methods on CIFAR100 (Krizhevsky et al., 2009) benchmark. We mark the best value in **bold**, and the second best with underline.

| Method | T0 | T1 | T2 | T3 | T4 | T5 | T6 | T7 | T8 | PD($\downarrow$) |
|---|---|---|---|---|---|---|---|---|---|---|
| SAVC (Song et al., 2023) | 78.77 | 73.31 | 69.31 | 64.93 | 61.70 | 59.25 | 57.13 | 55.19 | 53.12 | 25.65 |
| CaBD (Zhao et al., 2023) | 79.45 | 75.20 | 71.34 | 67.40 | 64.50 | 61.05 | 58.73 | 56.73 | 54.31 | 25.14 |
| TEEN (Wang et al., 2023) | 74.92 | 72.65 | 68.74 | 65.01 | 62.01 | 59.29 | 57.90 | 54.76 | 52.64 | 22.28 |
| OrCo (Ahmed et al., 2024) | 80.08 | 68.16 | 66.99 | 60.97 | 59.78 | 58.60 | 57.04 | 55.13 | 52.19 | 27.89 |
| Comp-FSCIL (Zou et al., 2024) | 80.93 | 76.52 | 72.69 | 68.52 | 65.50 | 62.62 | 60.96 | 59.27 | 56.71 | 24.22 |
| Bag of Tricks (Roy et al., 2024) | 80.25 | 77.20 | 75.09 | 70.82 | 67.83 | 64.86 | 62.73 | 60.52 | 58.55 | 21.70 |
| CLOSER (Oh et al., 2024) | 75.72 | 71.83 | 68.32 | 64.62 | 61.91 | 59.25 | 57.53 | 55.43 | 53.32 | 22.40 |
| Ours | 77.38 | 73.57 | 70.74 | 66.87 | 64.30 | 61.71 | 60.19 | 58.23 | 55.98 | **21.40** |

Table 4: Performance comparison with state-of-the-art methods on CUB200 (Wah et al., 2011) benchmark. We mark the best value in **bold**, and the second best with underline.

| Method | T0 | T1 | T2 | T3 | T4 | T5 | T6 | T7 | T8 | T9 | T10 | PD($\downarrow$) |
|---|---|---|---|---|---|---|---|---|---|---|---|---|
| SAVC (Song et al., 2023) | 81.85 | 77.92 | 74.95 | 70.21 | 69.96 | 67.02 | 66.16 | 65.30 | 63.84 | 63.15 | 62.50 | 19.35 |
| CaBD (Zhao et al., 2023) | 79.12 | 74.99 | 70.87 | 67.30 | 65.89 | 63.45 | 61.40 | 60.11 | 58.61 | 58.23 | 57.48 | 21.64 |
| TEEN (Wang et al., 2023) | 77.26 | 76.13 | 72.81 | 68.16 | 67.77 | 64.40 | 63.25 | 62.29 | 61.19 | 60.32 | 59.31 | 17.95 |
| OrCo (Ahmed et al., 2024) | 75.59 | 71.73 | 64.48 | 60.83 | 60.66 | 58.80 | 59.29 | 58.73 | 58.01 | 59.02 | 58.84 | 16.75 |
| Comp-FSCIL (Zou et al., 2024) | 80.94 | 77.51 | 74.34 | 71.00 | 68.77 | 66.41 | 64.85 | 63.92 | 62.12 | 62.10 | 61.17 | 19.77 |
| Bag of Tricks (Roy et al., 2024) | 82.31 | 78.03 | 75.45 | 70.99 | 71.06 | 67.85 | 67.44 | 66.05 | 64.95 | 64.31 | 63.60 | 18.71 |
| CLOSER (Oh et al., 2024) | 79.40 | 75.92 | 73.50 | 70.47 | 69.24 | 67.22 | 66.73 | 65.69 | 64.00 | 64.02 | 63.58 | 15.82 |
| Ours | 77.44 | 74.58 | 72.27 | 69.23 | 68.02 | 65.88 | 65.32 | 63.86 | 62.43 | 62.40 | 61.98 | **15.46** |

prototype regularization leads to more reliable predictions. Although the top-1 accuracies in the base session are nearly identical (80.77 for the baseline vs. 80.90 for ours), our approach consistently achieves higher base session accuracies for most values of $k$ (e.g., 95.70 for the baseline vs. 96.50 for our method at $k = 10$), a trend also observed in the top-1 last session accuracy (58.31 for the baseline vs. 59.96 for our method).

## 5.2 Further Analysis

**Hard-Negative Prototype Mining Analysis.** In this section, we investigate the impact of different hard-negative prototype mining approaches, which are explained in Section 4.1. For the static selection approach, we utilize a frozen ResNet pre-trained on the ImageNet dataset (Russakovsky et al., 2015) as the visual encoder and a frozen CLIP-ViT-B/32 (Radford et al., 2021) backbone as the textual encoder. As shown in Table 5, regardless of the selection approach, our proposed method improves performance compared to the baseline. Among the selection approaches, the dynamic selection approach generally demonstrates the best performance, likely due to its tailored adaptation capabilities that better accommodate shifting decision boundaries. Since the model's decision boundary shifts with each gradient step, the classes similar to a given instance may vary. While dominant hard-negative classes persist, the occasional inclusion of unusual hard-negative classes may help regularize the model by introducing inherent noise, thereby preventing overfitting to dominant classes during gradient estimation (Keskar et al., 2017). To verify this, we visualize the probabilities of selecting two classes as hard-negative pairs throughout training. We count the hard-negative classes for each anchor class per epoch at the sample level and aggregate these counts into probabilities after training. As shown in Figure 6, both approaches exhibit a similar trend (e.g., square patterns in miniImageNet) in selection, but the dynamic approach stochastically includes unusual classes – introducing variability that regularizes the model.

**Difficulty-Aware Prototype Regularization Analysis.** Our proposed approach focuses on increasing inter-class separation by pushing embeddings of similar (i.e., hard-negative) classes farther away. As shown in Table 6, we further conduct an experiment to demonstrate the effect of using prototypes of similar classes as hard-negatives in the FSCIL task. To this end, we compare ours with variant models with prototypes from (1) random classes and (2) dissimilar (i.e., easy-negative) classes. Note that, for the latter (2), we

Table 5: Performance comparison of different mining approaches for hard-negative prototype regularization (HNPR). The baseline method is trained using angular margin loss without HNPR.

| Method | HNPR | miniImageNet | | | CIFAR100 | | | CUB200 | | |
|---|---|---|---|---|---|---|---|---|---|---|
| | | Base | Last | PD($\downarrow$) | Base | Last | PD($\downarrow$) | Base | Last | PD($\downarrow$) |
| Baseline | | 79.30 | 57.62 | 21.68 | 77.37 | 54.55 | 22.82 | 77.30 | 60.86 | 16.44 |
| Static selection (visual) | ✓ | 80.15 | 58.89 | 21.26 | **78.02** | 55.35 | 22.67 | 76.71 | 60.72 | 15.99 |
| Static selection (textual) | ✓ | 79.97 | 58.82 | 21.15 | 77.03 | 54.40 | 22.63 | 76.75 | 60.72 | 16.03 |
| Dynamic selection (ours) | ✓ | **80.90** | **59.96** | **20.94** | 77.38 | **55.98** | **21.40** | **77.44** | **61.98** | **15.46** |

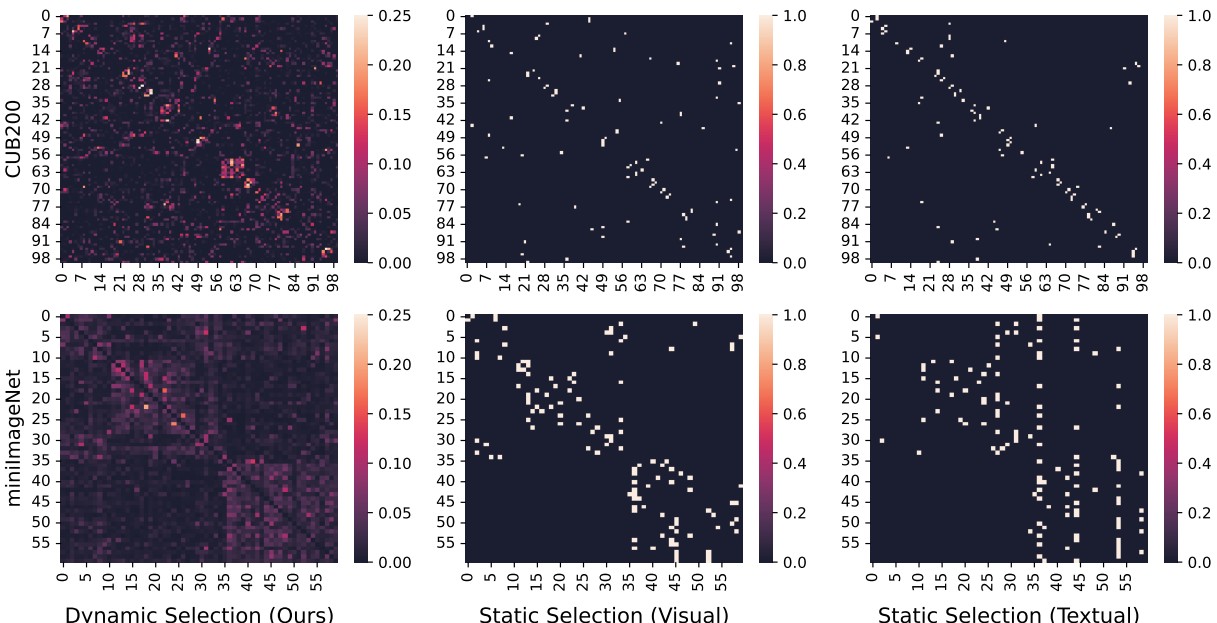

Figure 6: Visualization of the selection probabilities of two classes as hard-negative pairs during training, using different mining methods. The y-axis of each heatmap represents an anchor class, where we count the classes of hard-negative pairs (x-axis) per epoch. After training, we aggregate the counts of hard-negative pairs for each anchor class and normalize these accumulated counts to probabilities. Note that the overall trend in selection is similar across selection approaches, while the dynamic approach selects hard-negative pairs in a stochastic manner. Best viewed in color.

use our top-$k$ function in Section 4.1 but replace the argmax operation with argmin – choosing the most $k$ dissimilar classes excluding their ground-truth class. In Table 6, we observe that our prototype regularization with top-$k$ similar class mining indeed improves the overall FSCIL performance, clearly outperforming other design choices, i.e., use of randomly-chosen classes or use of top-$k$ dissimilar classes. Moreover, as we argued, the use of dissimilar classes in the FSCIL task is less significant in learning inter-class separation (it even performs worse than simply using randomly chosen classes as negative pairs). This may confirm that our approach to using hard-negative mining is effective in the FSCIL task. We further provide example images of the hard-negative and easy-negative classes corresponding to the anchor sample in the Appendix (see Figure A1).

**Ablation Studies.** We conduct an ablation study to assess the impact of elements composing our proposed method. As shown in Figure 7 (a), we compare our proposed method to other loss functions and show that our model outperforms the alternative loss functions including cross-entropy (CE) and angular margin (AM) loss. Further, in Figure 7 (b) and (c), we provide our analysis with different values of two main hyper-parameters, $k$ and $\hat{m}$. We observe that (1) models with $k = 2$ and $\hat{m} = 0.05$ show the best performance

Table 6: Performance comparison among different prototype regularization (PR) methods: (i) using random $k$ classes, (ii) using top-$k$ dissimilar classes (i.e., easy-negatives), and (iii) using top-$k$ similar classes (i.e., hard-negatives). The baseline method is trained using angular margin loss without PR.

| Method | PR | miniImageNet | | | CIFAR100 | | | CUB200 | | |
|---|---|---|---|---|---|---|---|---|---|---|
| | | Base | Last | PD($\downarrow$) | Base | Last | PD($\downarrow$) | Base | Last | PD($\downarrow$) |
| Baseline | | 79.30 | 57.62 | 21.68 | 77.37 | 54.55 | 22.82 | 77.30 | 60.86 | 16.44 |
| Random $k$ | ✓ | 80.07 | 58.55 | 21.52 | 77.75 | 55.57 | 22.18 | 77.44 | 61.41 | 16.03 |
| Easy-Negative top-$k$ | ✓ | 79.68 | 57.69 | 21.99 | **77.80** | 55.40 | 22.40 | **77.58** | 61.48 | 16.10 |
| Hard-Negative top-$k$ (ours) | ✓ | **80.90** | **59.96** | **20.94** | 77.38 | **55.98** | **21.40** | 77.44 | **61.98** | **15.46** |

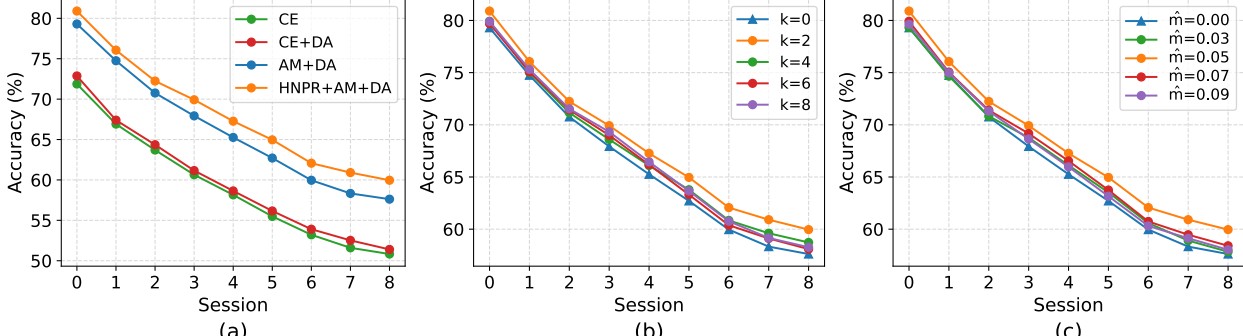

Figure 7: (a) Effect of data augmentation (DA) and comparison of various loss functions: cross-entropy loss (CE), angular margin loss (AM), and ours (HNPR). (b, c) Analysis of two hyperparameters in our proposed approach: (b) $k$ for the top-$k$ value, and (c) $\hat{m}$ for the penalty margin. A dynamic selection approach is applied to train all models on the miniImageNet dataset. Note that the blue line in (b) and (c) represents training without HNPR.

(see orange), and that (2) within a reasonable region of hyperparameters, HNPR clearly outperforms the baseline (compare circle vs. triangle).

## 6 Discussion

Batch-level hard-negative mining can lead to training instability, particularly under small batch sizes or label imbalance. This issue has been widely observed and reported in prior work on hard-negative mining (Yuan et al., 2022; Balmaseda et al., 2025). As our approach also relies on mining hard negatives within mini-batches, it inevitably inherits this limitation. In our experiments, we similarly observed signs of training instability when using small batch sizes (e.g., 128 for miniImageNet), which highlights a broader challenge inherent to hard-negative-based methods.

Despite this, one of the key advantages of our method lies in its computational efficiency. Although computational cost could also be a potential concern, we observe that our proposed method incurs only a negligible overhead compared to the baseline. As shown in Table 7, we measure the epoch runtime by executing both methods concurrently on the same device and averaging the results over five independent trials. This minimal cost arises because our method operates on class prototypes, adding only a small amount of computations during training.

Table 7: Comparison of average epoch runtime during the learning.

| Method | CIFAR100 | miniImageNet |
|---|---|---|
| Baseline | 46.02s±3.26 | 107.16s±4.27 |
| Ours | 46.12s±3.60 | 107.19s±4.18 |

## 7 Conclusion

In this work, we propose a novel method for imposing a penalty margin between the sample and the hard-negative prototypes, which effectively guide the model to focus on the distinctions between challenging classes. We first analyze the impact of difficulty variation of data, demonstrating that hard-negative samples

and classes are key factors for effective FSCIL. Based on this observation, we design a hard-negative prototype regularization approach and explore two distinct selection strategies for these prototypes. Extensive experiments and analyses on widely used benchmarks demonstrate the superiority of our proposed method, achieving state-of-the-art performance.

**Acknowledgements.** This work was supported by IITP grant funded by the Korea government(MSIT) (IITP-2025-RS-2024-00397085, 30%, RS-2025-02263754, IITP-2025-RS-2025-02304828, 10%, RS-2022-II220043, Adaptive Personality for Intelligent Agents, 30%). This work was also supported by the Korea Institute of Science and Technology (KIST) Institutional Program (Project No. 2E33611, 30%).

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

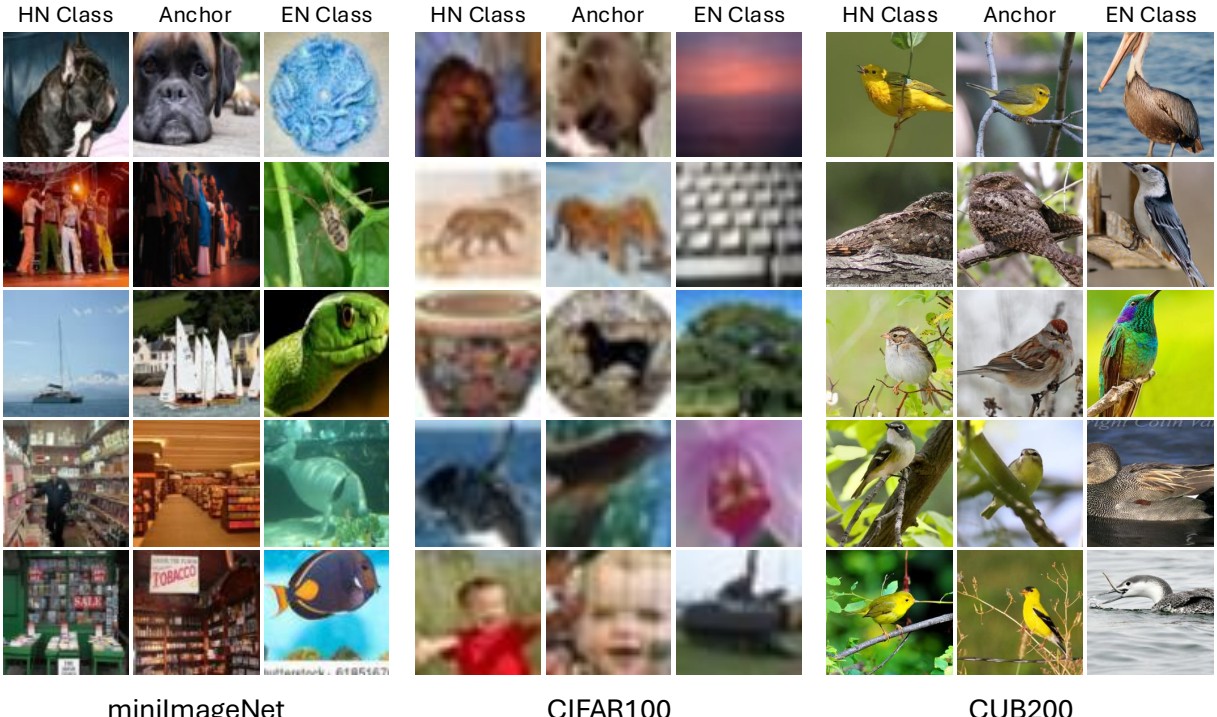

Figure A1: Example images of the classes selected as hard-negative (HN) and easy-negative (EN) relative to the anchor sample, across three widely used datasets in few-shot class-incremental learning (FSCIL): miniImageNet (Vinyals et al., 2016), CIFAR100 (Krizhevsky et al., 2009), and CUB200 (Wah et al., 2011). Note the strong visual similarity between the hard-negative class and anchor images, and the clear visual dissimilarity between the easy-negative class and anchor images.

## A    Appendix

### A.1    Further Experimental Results

In this section, we provide additional qualitative and quantitative results that further support our technical claims made in the main paper.

**Loss Ablation Study.** Our proposed approach, defined in Equation (8) in the main paper, consists of three main loss terms in the denominator: intra-class cohesion ($\mathcal{L}_A$, with $\theta_{y_i}$), inter-class separation ($\mathcal{L}_B$, with $\theta_j$), and regularized inter-class separation with hard-negative prototype mining ($\mathcal{L}_C$, with $\theta_c$). We perform an ablation study to evaluate the impact of each loss term. As shown in Table A1, we observe that (i) the model trained with all three loss terms achieves the best performance (compare the last row with others), and (ii) regularized inter-class separation with hard-negative prototype mining significantly improves performance when combined with the inter-class separation loss term. This improvement likely occurs because combining these two terms provides both global and local guidance, enabling the model to better align embeddings across different classes.

Table A1: Performance comparison between variant models trained with different loss terms in our method. $\mathcal{L}_A$: Intra-class cohesion, $\mathcal{L}_B$: Inter-class separation, and $\mathcal{L}_C$: Regularized inter-class separation with hard-negative prototype mining.

| $\mathcal{L}_A$ | $\mathcal{L}_B$ | $\mathcal{L}_C$ | Base | Last | PD($\downarrow$) |
|---|---|---|---|---|---|
| ✓ | ✓ | | 79.30 | 57.62 | 21.68 |
| ✓ | | ✓ | 78.60 | 57.20 | 21.40 |
| ✓ | ✓ | ✓ | 80.90 | 59.96 | **20.94** |

**Negative Class Examples.** To qualitatively evaluate our proposed method, we visualize example images from hard-negative and easy-negative classes relative to the anchor sample, using cosine distance as the

Table A2: Performance comparison with ViT-based state-of-the-art method, Attention-aware Self-adaptive Prompt (ASP) (Liu et al., 2024), on ImageNet-R (Hendrycks et al., 2021) benchmark. AM denotes angular margin loss. We mark the best value in **bold**.

| Method | T0 | T1 | T2 | T3 | T4 | T5 | T6 | T7 | T8 | T9 | T10 | PD($\downarrow$) |
|---|---|---|---|---|---|---|---|---|---|---|---|---|
| ASP | 81.34 | 78.46 | 77.96 | 74.72 | 73.72 | 72.73 | 71.05 | 70.29 | 69.98 | 68.58 | 67.07 | 14.27 |
| ASP+AM | **81.64** | 78.49 | 77.88 | 75.08 | 73.70 | 72.76 | 71.40 | 70.89 | 70.24 | 69.43 | 68.27 | 13.37 |
| ASP+Ours | 81.38 | **78.81** | **78.60** | **75.54** | **74.50** | **73.64** | **72.12** | **71.22** | **70.51** | **69.46** | **68.55** | **12.83** |

Table A3: Mean and standard deviation across five independent trials with different random seeds on the miniImageNet benchmark. AM denotes the angular margin loss. We reproduce the TEEN (Wang et al., 2023) method using the official code repository.

| Method | T0 | T1 | T2 | T3 | T4 | T5 | T6 | T7 | T8 | PD($\downarrow$) |
|---|---|---|---|---|---|---|---|---|---|---|
| TEEN | 75.14±0.29 | 70.27±0.45 | 68.29±0.44 | 63.14±0.49 | 60.45±0.48 | 57.70±0.52 | 55.00±0.50 | 53.23±0.54 | 51.83±0.52 | 23.31±0.43 |
| AM | 79.08±0.40 | 75.03±0.49 | 71.19±0.40 | 68.56±0.53 | 65.78±0.48 | 63.07±0.53 | 60.26±0.36 | 58.62±0.37 | 57.78±0.27 | 21.91±0.16 |
| Ours | **79.96±0.56** | **75.36±0.50** | **71.51±0.52** | **68.93±0.75** | **66.31±0.74** | **63.74±0.82** | **60.91±0.71** | **59.62±0.73** | **58.73±0.70** | **21.24±0.30** |

similarity metric. As shown in Figure A1, images from hard-negative classes tend to be visually similar to the anchor samples, whereas those from easy-negative classes are noticeably different from the anchor images. These examples illustrate that our method effectively identifies challenging negative classes, highlighting its capability to enhance discriminative learning during model training.

**ViT-Based FSCIL Evaluation.** Recent work has begun to leverage the generalization capabilities of modern pre-trained Vision Transformer (ViT) (Dosovitskiy et al., 2020) models to FSCIL. To demonstrate the effectiveness of our approach on such architectures, we adopt the state-of-the-art ViT-based FSCIL method— Attention-aware Self-adaptive Prompt (ASP) (Liu et al., 2024)—as our baseline and directly compare its performance to ours. Following the ASP evaluation protocol, we use the ImageNet-R (Hendrycks et al., 2021) benchmark, a real-world distribution-shift dataset not previously explored in this context. As shown in the Table A2, our hard-negative-based prototype regularization outperforms both vanilla ASP and ASP enhanced with an angular margin (AM). Note that we use a much larger ViT backbone (176M parameters) compared to ResNet-18 (20M), which makes our gains even more compelling.

**Multi-Run Evaluation on miniImageNet.** To validate the effectiveness of our proposed approach, we conducted five independent trials with different random seeds on the miniImageNet benchmark. We compare three different methods: (i) TEEN (Wang et al., 2023), the previous state-of-the-art method; (ii) the baseline (angular margin), as described in Section 3.1.; and (iii) our proposed method, hard-negative-based prototype regularization. As shown in the Table A3, our method consistently achieves the best performance among the three. Note that we reproduced the TEEN method using its official code repository.

**Comparison with Sample-Wise Hard-Example Mining Method.** To quantify the benefits of our prototype-based hard-negative prototype regularization (HNPR) over a conventional sample-wise hard example mining (HEM) approach, we compare against the standard triplet-loss (Weinberger & Saul, 2009) framework on the CUB200 benchmark under the FSCIL protocol. Table A4 presents the session-wise accuracy of both methods: our HNPR consistently outperforms the triplet-loss baseline, demonstrating superior mitigation of class confusion and forgetting by leveraging class prototypes rather than individual hard samples.

**Extended Tables.** We provide the complete tables comparing our proposed method against state-of-the-art approaches on the CIFAR100 and CUB200 benchmark. Note that Tables 3 and 4 in the manuscript are the shorter version of Tables A5 and A6.

Table A4: Performance comparison with sample-wise hard-negative mining method, triplet loss (Weinberger & Saul, 2009) with different margin ($m$) values on CUB200 benchmark.

| Method | T0 | T1 | T2 | T3 | T4 | T5 | T6 | T7 | T8 | T9 | T10 | PD($\downarrow$) |
|---|---|---|---|---|---|---|---|---|---|---|---|---|
| Triplet ($m = 0.3$) | 76.26 | 73.34 | 71.14 | 67.84 | 66.31 | 64.45 | 63.29 | 62.27 | 60.70 | 60.76 | 60.11 | 16.15 |
| Triplet ($m = 0.4$) | 76.47 | 73.59 | 71.52 | 68.08 | 66.34 | 64.49 | 63.24 | 62.25 | 60.53 | 60.63 | 59.91 | 16.56 |
| Ours | **77.44** | **74.58** | **72.27** | **69.23** | **68.02** | **65.88** | **65.32** | **63.86** | **62.43** | **62.40** | **61.98** | **15.46** |

Table A5: Performance comparison with state-of-the-art methods on CIFAR100 (Krizhevsky et al., 2009) benchmark. We mark the best value in **bold**, and the second best with underline.

| Method | T0 | T1 | T2 | T3 | T4 | T5 | T6 | T7 | T8 | PD($\downarrow$) |
|---|---|---|---|---|---|---|---|---|---|---|
| TOPIC (Tao et al., 2020) | 64.10 | 55.88 | 47.07 | 45.16 | 40.11 | 36.38 | 33.96 | 31.55 | 29.37 | 34.73 |
| CLOM (Zou et al., 2022) | 74.20 | 69.83 | 66.17 | 62.39 | 59.26 | 56.48 | 54.36 | 52.16 | 50.25 | 23.95 |
| ALICE (Peng et al., 2022) | 79.00 | 70.50 | 67.10 | 63.40 | 61.20 | 59.20 | 58.10 | 56.30 | 54.10 | 24.90 |
| FACT (Zhou et al., 2022a) | 74.60 | 72.09 | 67.56 | 63.52 | 61.38 | 58.36 | 56.28 | 54.24 | 52.10 | 22.50 |
| LIMIT (Zhou et al., 2022b) | 73.81 | 72.09 | 67.87 | 63.89 | 60.70 | 57.77 | 55.67 | 53.52 | 51.23 | 22.58 |
| GKEAL (Zhuang et al., 2023) | 74.01 | 70.45 | 67.01 | 63.08 | 60.01 | 57.30 | 55.50 | 53.39 | 51.40 | 22.61 |
| WaRP (Kim et al., 2023) | 80.31 | 75.86 | 71.87 | 67.58 | 64.39 | 61.34 | 59.15 | 57.10 | 54.74 | 25.57 |
| SoftNet (Kang et al., 2023) | 79.88 | 75.54 | 71.64 | 67.47 | 64.45 | 61.09 | 59.07 | 57.29 | 55.33 | 24.55 |
| SAVC (Song et al., 2023) | 78.77 | 73.31 | 69.31 | 64.93 | 61.70 | 59.25 | 57.13 | 55.19 | 53.12 | 25.65 |
| CaBD (Zhao et al., 2023) | 79.45 | 75.20 | 71.34 | 67.40 | 64.50 | 61.05 | 58.73 | 56.73 | 54.31 | 25.14 |
| TEEN (Wang et al., 2023) | 74.92 | 72.65 | 68.74 | 65.01 | 62.01 | 59.29 | 57.90 | 54.76 | 52.64 | 22.28 |
| OrCo (Ahmed et al., 2024) | 80.08 | 68.16 | 66.99 | 60.97 | 59.78 | 58.60 | 57.04 | 55.13 | 52.19 | 27.89 |
| Comp-FSCIL (Zou et al., 2024) | 80.93 | 76.52 | 72.69 | 68.52 | 65.50 | 62.62 | 60.96 | 59.27 | 56.71 | 24.22 |
| Bag of Tricks (Roy et al., 2024) | 80.25 | 77.20 | 75.09 | 70.82 | 67.83 | 64.86 | 62.73 | 60.52 | 58.55 | 21.70 |
| CLOSER (Oh et al., 2024) | 75.72 | 71.83 | 68.32 | 64.62 | 61.91 | 59.25 | 57.53 | 55.43 | 53.32 | 22.40 |
| Ours | 77.38 | 73.57 | 70.74 | 66.87 | 64.30 | 61.71 | 60.19 | 58.23 | 55.98 | **21.40** |

## A.2 Implementation Details

**Hyperparameters.** For all experiments, we use a ResNet-18 (RN18) backbone with a projection head of dimension 2048. The number of negative pairs $k$ varies depending on the dataset: we use $k = 1$ for CUB200, and $k = 2$ for both CIFAR100 and miniImageNet. We train our model using an SGD optimizer with the following learning rates: 3e-2 for miniImageNet, 6e-2 for CIFAR100, and 3e-3 for CUB200. Cosine scheduling with warm-up is employed for learning rate scheduling. For the warm-up, we utilize 3% of the total epochs. The models are trained for 80 epochs on CUB200, 100 epochs on CIFAR100, and 300 epochs on miniImageNet, with batch sizes of 1024, 512, and 512, respectively.

**Data Augmentation.** We utilize the augmentation strategies frequently employed in prior FSCIL research (Yang et al., 2023; Peng et al., 2022; Zhang et al., 2021; Tao et al., 2020): (1) resized cropping, (2) horizontal flipping, and (3) color jittering. Given the variation in the size of input images, we adjust the size of the resized crop according to the dataset (224 for CUB200, 32 for CIFAR100, and 84 for miniImageNet). Additionally, horizontal flipping is applied to the cropped images with a probability of 0.5. For color jittering, we establish a parameter of 0.4 for the adjustment of brightness, contrast, and saturation.

Table A6: Performance comparison with state-of-the-art methods on CUB200 (Wah et al., 2011) benchmark. We mark the best value in **bold**, and the second best with underline.

| Method | T0 | T1 | T2 | T3 | T4 | T5 | T6 | T7 | T8 | T9 | T10 | PD($\downarrow$) |
|---|---|---|---|---|---|---|---|---|---|---|---|---|
| TOPIC (Tao et al., 2020) | 68.68 | 62.49 | 54.81 | 49.99 | 45.25 | 41.40 | 38.35 | 35.36 | 32.22 | 28.31 | 26.28 | 42.40 |
| CLOM (Zou et al., 2022) | 79.57 | 76.07 | 72.94 | 69.82 | 67.80 | 65.56 | 63.94 | 62.59 | 60.62 | 60.34 | 59.58 | 19.99 |
| ALICE (Peng et al., 2022) | 77.40 | 72.70 | 70.60 | 67.20 | 65.90 | 63.40 | 62.90 | 61.90 | 60.50 | 60.60 | 60.10 | 17.30 |
| FACT (Zhou et al., 2022a) | 75.90 | 73.23 | 70.84 | 66.13 | 65.56 | 62.15 | 61.74 | 59.83 | 58.41 | 57.89 | 56.94 | 18.96 |
| LIMIT (Zhou et al., 2022b) | 76.32 | 74.18 | 72.68 | 69.19 | 68.79 | 65.64 | 63.57 | 62.69 | 61.47 | 60.44 | 58.45 | 17.87 |
| GKEAL (Zhuang et al., 2023) | 78.88 | 75.62 | 72.32 | 68.62 | 67.23 | 64.26 | 62.98 | 61.89 | 60.20 | 59.21 | 58.67 | 20.21 |
| WaRP (Kim et al., 2023) | 77.74 | 74.15 | 70.82 | 66.90 | 65.01 | 62.64 | 61.40 | 59.86 | 57.95 | 57.77 | 57.01 | 20.73 |
| SoftNet (Kang et al., 2023) | 78.07 | 74.58 | 71.37 | 67.54 | 65.37 | 62.60 | 61.07 | 59.37 | 57.53 | 57.21 | 56.75 | 21.32 |
| SAVC (Song et al., 2023) | 81.85 | 77.92 | 74.95 | 70.21 | 69.96 | 67.02 | 66.16 | 65.30 | 63.84 | 63.15 | 62.50 | 19.35 |
| CaBD (Zhao et al., 2023) | 79.12 | 74.99 | 70.87 | 67.30 | 65.89 | 63.45 | 61.40 | 60.11 | 58.61 | 58.23 | 57.48 | 21.64 |
| TEEN (Wang et al., 2023) | 77.26 | 76.13 | 72.81 | 68.16 | 67.77 | 64.40 | 63.25 | 62.29 | 61.19 | 60.32 | 59.31 | 17.95 |
| OrCo (Ahmed et al., 2024) | 75.59 | 71.73 | 64.48 | 60.83 | 60.66 | 58.80 | 59.29 | 58.73 | 58.01 | 59.02 | 58.84 | 16.75 |
| Comp-FSCIL (Zou et al., 2024) | 80.94 | 77.51 | 74.34 | 71.00 | 68.77 | 66.41 | 64.85 | 63.92 | 62.12 | 62.10 | 61.17 | 19.77 |
| Bag of Tricks (Roy et al., 2024) | 82.31 | 78.03 | 75.45 | 70.99 | 71.06 | 67.85 | 67.44 | 66.05 | 64.95 | 64.31 | 63.60 | 18.71 |
| CLOSER (Oh et al., 2024) | 79.40 | 75.92 | 73.50 | 70.47 | 69.24 | 67.22 | 66.73 | 65.69 | 64.00 | 64.02 | 63.58 | 15.82 |
| Ours | 77.44 | 74.58 | 72.27 | 69.23 | 68.02 | 65.88 | 65.32 | 63.86 | 62.43 | 62.40 | 61.98 | **15.46** |

