# OpenReview forum: "Hard-Negative Prototype-Based Regularization for Few-Shot Class-Incremental Learning"
_TMLR — Accepted by TMLR_

### Review · Reviewer_4yva · 2025-06-25

**Summary Of Contributions:**

The authors propose a hard-negative prototype regularization approach which imposes a penalty margin between samples and their similar class prototypes.  They explore dynamic and static training selection strategies.  The approach is validated on miniImageNet, CIFAR100, and CUB200.

**Audience:**

Yes

**Claims And Evidence:**

No

**Requested Changes:**

Please run repeat trials on at least one of the datasets (Tables 1,3,4) so we can see the significance of the approaches.

Please address any compute considerations which may make the proposed method advantageous/disadvantageous, especially compared to other methods.  Are there requirements around memory, compute time, total number of classes, etc.?  Speaking to limitations of the current method is also important in general.

I think demonstrating results on a more modern model, much larger than ResNet, is important for demonstrating usefulness of the approach.

**Strengths And Weaknesses:**

STRENGTHS:
The approach is straightforward and well motivated.

The results are on par with other methods, although I would disagree with the claim that the method has been demonstrated to be superior than the others.

The writing is clear and easy to follow.

WEAKNESSES:
While the application to FSCIL may be novel, the importance and use of hard example learning for few shot learning is well established.  It would be helpful to articulate early on why hard example learning within FSCIL is sufficiently novel and impactful as opposed to an application of a standard framework (HEM) to a specific newer method (FSCIL).  Grounding this approach in other HEM works applied to other few shot methods would help with this.  Or for example, a sentence like "We... observe that hard-negative samples and classes are key to success in FSCIL"- this observation is seen in other few shot approaches so it would be helpful to articulate (and cite) that.  Works within metric learning are referenced, but expanding to the broader one-shot and few-shot learning literature would be helpful.

The datasets selected are relatively small by today's standards.  Additionally, they are all classification focused whereas much of the interest in FSL approaches is for detection and segmentation.  While these are the same datasets that were used in FSCIL, that work was more theoretically driven/motivated so these smaller, simpler, toy datasets are more justified.

Similarly all experiments are conducted using Resnet-18, which is a very small model by today's standards.  For theoretical proof, a small model is fine (i.e. to demonstrate that a method produces correct results).  However, to demonstrate the usefulness of a method, particularly something like this where the primary application would be in large scale, real-world datasets, likely leveraging substantially larger models, using a more relevant model would make more sense.  ResNet-18 and toy datasets establish that the approach works, but something more substantive is needed to demonstrate that the approach is useful for this particular task.

The results in Table 2 suggest that while negative samples, especially hard-negative samples, lead to improvement (a result which has been seen with other approaches in metric learning, HEM, etc.), the baseline and proposed approaches are not meaningfully different.

Error bars are provided with Table 2, but not elsewhere.  Changing the classes which are novel, or changing which specific samples are used could be done to show significance on the other results as well.

---

> ### Author Response · Authors · 2025-08-08
> **Author Response (1/3)**
>
> We thank Reviewer 4yva (R1) for the detailed and valuable feedback. Below, we summarize the weaknesses (W) and requested changes (RC) under the following headings:
> - Justification of the novelty in FSCIL (W1)
> - Validation on modern datasets and models (W2, RC3)
> - Effectiveness of the approach (W3, RC1)
> - Computational considerations (RC2)
>
> **Justification of the novelty in FSCIL (W1)**
> > "**(W1)** While the application to FSCIL may be novel, the importance and use of hard example learning for few shot learning is well established. It would be helpful to articulate early on why hard example learning within FSCIL is sufficiently novel and impactful as opposed to an application of a standard framework (HEM) to a specific newer method (FSCIL). Grounding this approach in other HEM works applied to other few shot methods would help with this."
>
> This is a great suggestion. To the best of our knowledge, the effectiveness of applying hard example mining (HEM) to FSCIL—which integrates few-shot learning (FSL) and class-incremental learning (CIL)—has not yet been explored, even though prior FSL studies have demonstrated the importance and utility of HEM [1-3]. Therefore, in Section 3, we first analyze the role of hard examples (and classes) in the FSCIL setting, and show that hard examples are also critical in CIL scenarios, as they are a primary cause of confusion and forgetting. Based on this observation, we leverage prototypes—commonly used in FSCIL methods—to regularize samples using hard-negative concepts.
>
> To compare our method with standard sample-wise HEM framework, triplet loss [4], we have conducted an experiment on the CUB200 benchmark. As shown in the table below, our prototype-based HNPR approach consistently outperforms the standard HEM method in the FSCIL scenario.
> | Method |   T0   |   T1   |   T2   |   T3   |   T4   |   T5   |   T6   |   T7   |   T8   |   T9   |  T10   | PD (↓) |
> |-------------------|--------|--------|--------|--------|--------|--------|--------|--------|--------|--------|--------|--------|
> | Triplet (m=0.3)   |  76.26 |  73.34 |  71.14 |  67.84 |  66.31 |  64.45 |  63.29 |  62.27 |  60.70 |  60.76 |  60.11 |  16.15 |
> | Triplet (m=0.4)   |  76.47 |  73.59 |  71.52 |  68.08 |  66.34 |  64.49 |  63.24 |  62.25 |  60.53 |  60.63 |  59.91 |  16.56 |
> | **Ours**          | **77.44** | **74.58** | **72.27** | **69.23** | **68.02** | **65.88** | **65.32** | **63.86** | **62.43** | **62.40** | **61.98** | **15.46** |

---

> ### Author Response · Authors · 2025-08-08
> **Author Response (2/3)**
>
> **Validation on Modern Datasets and Models (W2, RC3)**
> > "**(W2)** The datasets selected are relatively small by today's standards. Additionally, they are all classification focused whereas much of the interest in FSL approaches is for detection and segmentation. While these are the same datasets that were used in FSCIL, that work was more theoretically driven/motivated so these smaller, simpler, toy datasets are more justified.
> > Similarly all experiments are conducted using Resnet-18, which is a very small model by today's standards. For theoretical proof, a small model is fine (i.e. to demonstrate that a method produces correct results). However, to demonstrate the usefulness of a method, particularly something like this where the primary application would be in large scale, real-world datasets, likely leveraging substantially larger models, using a more relevant model would make more sense. ResNet-18 and toy datasets establish that the approach works, but something more substantive is needed to demonstrate that the approach is useful for this particular task.
> > ⠀
> > **(RC3)** I think demonstrating results on a more modern model, much larger than ResNet, is important for demonstrating the usefulness of the approach."
>
> This is an excellent suggestion. Recent work has begun to leverage the generalization capabilities of modern pre-trained Vision Transformer (ViT) models to FSCIL [5-7]. To demonstrate the effectiveness of our approach on such architectures, we adopt the state-of-the-art ViT-based FSCIL method—Attention-aware Self-adaptive Prompt (ASP)—as our baseline and directly compare its performance to ours. Following the ASP evaluation protocol, we use the ImageNet-R benchmark, a real-world distribution-shift dataset not previously explored in this context. As shown in the table below, our hard-negative-based prototype regularization outperforms both vanilla ASP and ASP enhanced with an angular margin (AM). Note that we use a much larger ViT backbone (176M parameters) compared to ResNet-18 (20M), which makes our gains even more compelling.
> | Method    |   T0    |   T1    |   T2    |   T3    |   T4    |   T5    |   T6    |   T7    |   T8    |   T9    |   T10   | PD (↓)  |
> |-----------|---------|---------|---------|---------|---------|---------|---------|---------|---------|---------|---------|---------|
> | ASP       |   81.34 |   78.46 |   77.96 |   74.72 |   73.72 |   72.73 |   71.05 |   70.29 |   69.98 |   68.58 |   67.07 |   14.27 |
> | ASP+AM    | **81.64** |   78.49 |   77.88 |   75.08 |   73.70 |   72.76 |   71.40 |   70.89 |   70.24 |   69.43 |   68.27 |   13.37 |
> | ASP+Ours  |   81.38 | **78.81** | **78.60** | **75.54** | **74.50** | **73.64** | **72.12** | **71.22** | **70.51** | **69.46** | **68.55** | **12.83** |
>
> **Effectiveness of the Approach (W3, RC1)**
> > "**(W3)** The results in Table 2 suggest that while negative samples, especially hard-negative samples, lead to improvement (a result which has been seen with other approaches in metric learning, HEM, etc.), the baseline and proposed approaches are not meaningfully different. Error bars are provided with Table 2, but not elsewhere. Changing the classes which are novel, or changing which specific samples are used could be done to show significance on the other results as well.
> > ⠀
> > **(RC1)** Please run repeat trials on at least one of the datasets (Tables 1,3,4) so we can see the significance of the approaches.”
>
> Thank you for the important feedback. To validate the effectiveness of our proposed approach, we conducted five independent trials with different random seeds on the miniImageNet benchmark. We compare three different methods: (i) TEEN, the previous state-of-the-art method; (ii) the baseline (angular margin), as described in the section 3.1.; and (iii) our proposed method, hard-negative-based prototype regularization. As shown in the table below, our method consistently achieves the best performance among the three. Note that we reproduced the TEEN method using its official code repository.
>
> | Method   |     T0      |     T1      |     T2      |     T3      |     T4      |     T5      |     T6      |     T7      |     T8      |   PD (↓)    |
> |----------|-------------|-------------|-------------|-------------|-------------|-------------|-------------|-------------|-------------|-------------|
> | TEEN     | 75.14±0.29  | 70.27±0.45  | 66.29±0.44  | 63.14±0.49  | 60.45±0.48  | 57.70±0.52  | 55.00±0.50  | 53.23±0.54  | 51.83±0.52  | 23.31±0.43  |
> | Baseline | 79.69±0.40  | 75.03±0.49  | 71.19±0.40  | 68.56±0.53  | 65.78±0.48  | 63.07±0.53  | 60.26±0.36  | 58.62±0.37  | 57.78±0.27  | 21.91±0.16  |
> | **Ours** | **79.96±0.56** | **75.35±0.50** | **71.51±0.52** | **68.93±0.75** | **66.31±0.74** | **63.74±0.82** | **60.91±0.71** | **59.62±0.73** | **58.73±0.70** | **21.24±0.30** |

---

> ### Author Response · Authors · 2025-08-08
> **Author Response (3/3)**
>
> **Computational Considerations (RC2)**
> > “**(RC2)** Please address any compute considerations which may make the proposed method advantageous/disadvantageous, especially compared to other methods. Are there requirements around memory, compute time, total number of classes, etc.? Speaking to limitations of the current method is also important in general.”
>
> To assess the computational overhead of our proposed method, we measured the epoch runtime and compared it to the baseline. For a fair comparison, both methods (i.e., baseline and ours) were executed simultaneously on the same device, and the results were averaged over five independent trials.
>
> | Method   | CIFAR100     | miniImageNet    |
> |----------|--------------|-----------------|
> | Baseline | 46.02s±3.26  | 107.16s±4.27    |
> | Ours     | 46.12s±3.60  | 107.19s±4.18    |
>
> As shown in the table above, our hard-negative prototype regularization introduces only a negligible computational overhead compared to the baseline.  This minimal cost is due to the fact that our method operates on class prototypes, introducing only lightweight additional computations during training. We have added the discussion section including this result in our manuscript.
>
>
> **References**
> > [1] Mandalika, Sriram. "Generalizable Vision-Language Few-Shot Adaptation with Predictive Prompts and Negative Learning." arXiv preprint arXiv:2505.11758 (2025)
> > [2] Roy, Aniket, et al. "Felmi: Few shot learning with hard mixup." Advances in Neural Information Processing Systems 35 (2022): 24474-24486.
> > [3] Li, Yiting, et al. "Few-shot object detection via classification refinement and distractor retreatment." Proceedings of the IEEE/CVF conference on computer vision and pattern recognition. 2021.
> > [4] Weinberger, Kilian Q., and Lawrence K. Saul. "Distance metric learning for large margin nearest neighbor classification." Journal of machine learning research 10.2 (2009).
> > [5] Huang, Zitong, et al. "Learning prompt with distribution-based feature replay for few-shot class-incremental learning." arXiv preprint arXiv:2401.01598 (2024).
> > [6] Park, Keon-Hee, Kyungwoo Song, and Gyeong-Moon Park. "Pre-trained vision and language transformers are few-shot incremental learners." Proceedings of the IEEE/CVF Conference on Computer Vision and Pattern Recognition. 2024.
> > [7] Liu, Chenxi, et al. "Few-shot class incremental learning with attention-aware self-adaptive prompt." European Conference on Computer Vision. Cham: Springer Nature Switzerland, 2024.

---

### Review · Reviewer_XGo1 · 2025-07-14

**Summary Of Contributions:**

This paper proposes Hard-Negative Prototype Regularization (HNPR) for FSCIL. The method adds an extra angular margin between a sample and its most similar (hard-negative) class prototypes to reduce confusion. Two mining strategies are explored, dynamic (from model predictions) and static (from pretrained encoders).

**Audience:**

Yes

**Claims And Evidence:**

Yes

**Requested Changes:**

1. Clarify novelty: The use of hard-negative mining is well-established in metric learning and has appeared in FSCIL. The paper must clearly state what is actually new here. e.g., prototype-level margin within the FSCIL angular-margin framework, and avoid overstating “first” claims.

2. Improve related work discussion: Expand the related work section to properly acknowledge prior literature on hard-negative mining in metric learning, few-shot learning, and FSCIL. Show how your method differs technically and conceptually.

3.Discuss potential drawbacks of dynamic mining: Batch-level hard-negative selection could introduce noise or instability. Include a discussion (or experiment) on whether this affects training dynamics, especially under small batch sizes or label imbalance.

**Strengths And Weaknesses:**

**Strengths:**
1. Clean idea, easy to implement. The method just adds one more margin term into a known loss (angular margin loss), so it doesn’t require new architectures or fancy backbones.


**Weakness:**
1. **Motivation is overstated.** The idea of focusing on hard-negative examples is far from new. It’s a well-established concept in metric learning, face recognition (e.g., FaceNet, ArcFace), and even in some FSCIL methods like[3] [4] that implicitly push apart similar classes.
Also, related work section lists metric-learning papers but does not discuss why their hard-negative strategies cannot be directly applied , weakening the novelty argument.

[1]FaceNet: A Unified Embedding for Face Recognition and Clustering (CVPR 2015)

[2] ArcFace: Additive Angular Margin Loss for Deep Face Recognition

[3] Learning with Fantasy: Semantic‑Aware Virtual Contrastive Constraint for Few‑Shot Class‑Incremental Learning (CVPR 2023)

[4] Rethinking Few‑shot Class‑incremental Learning (ECCV 2024)

2. Adding a uniform extra margin for all hard-negative classes ignores that some pairs may already be well-separated, potentially over-penalising and harming calibration. No analysis of failure cases or calibration metrics is given.

---

> ### Author Response · Authors · 2025-08-08
> **Author Response (1/2)**
>
> We appreciate Reviewer XGo1 (R2) for the detailed and valuable comment. The weaknesses (W) and requested changes (RC) are summarized below under these headings:
> - Novelty clarification (W1, RC1, RC2)
> - Failure case analysis and potential drawbacks (W2, RC3)
>
> **Novelty Clarification (W1, RC1, RC2)**
> > “**(W1)** Motivation is overstated. The idea of focusing on hard-negative examples is far from new. It’s a well-established concept in metric learning, face recognition (e.g., FaceNet, ArcFace), and even in some FSCIL methods like[3][4] that implicitly push apart similar classes. Also, the related work section lists metric-learning papers but does not discuss why their hard-negative strategies cannot be directly applied, weakening the novelty argument.
> > [1] FaceNet: A Unified Embedding for Face Recognition and Clustering (CVPR 2015)
> > [2] ArcFace: Additive Angular Margin Loss for Deep Face Recognition
> > [3] Learning with Fantasy: Semantic‑Aware Virtual Contrastive Constraint for Few‑Shot Class‑Incremental Learning (CVPR 2023)
> > [4] Rethinking Few‑shot Class‑incremental Learning (ECCV 2024)
> > ⠀
> > **(RC1)** Clarify novelty: The use of hard-negative mining is well-established in metric learning and has appeared in FSCIL. The paper must clearly state what is actually new here. e.g., prototype-level margin within the FSCIL angular-margin framework, and avoid overstating “first” claims.
> > ⠀
> > **(RC2)** Improve related work discussion: Expand the related work section to properly acknowledge prior literature on hard-negative mining in metric learning, few-shot learning, and FSCIL. Show how your method differs technically and conceptually.”
>
> Thank you for the feedback and the opportunity to clarify our novelty. We agree that hard-negative mining is well-established concept in metric learning [5-7]. However, we argue that our work is still novel with respect to (i) clarification of the importance of hard examples in FSCIL—which has not yet been explored in the community—, and (ii) effective prototype regularization method that leverages hard-negatives for FSCIL. We will compare the difference to given FSCIL references [3, 4] and further clarify our novelty in the following paragraphs.
>
> **Ours vs. SAVC [3].** SAVC (Semantic-Aware Virtual Contrastive) generates virtual classes by augmenting base classes through pre-defined transformation, and employs supervised contrastive learning to boost base class separation and novel class generalization. Although these virtual classes may be viewed as ‘potentially hard’ classes, its primary goal is to leverage the rich semantic variations by the virtual classes—not to model or emphasize the discriminative difficulty of individual samples or classes.
>
> **Ours vs. YourSelf [4].** YourSelf [4] leverages intermediate layer features to refine the final representations by aligning their distributions. Similar to SAVC, its primary goal is to leverage more generalizable features to improve performance on unseen classes. However, the method does not consider the varying levels of discriminative difficulty across samples and classes.
>
> **Novelty Clarification.** In contrast, as discussed in Section 3, we first explore how difficult examples and classes impact the FSCIL setting, and demonstrate that these “hard” cases play a pivotal role by including confusion and catastrophic forgetting. Motivated by this observation, we utilize prototypes—commonly adopted in FSCIL methods—to regularize the learning process through hard-negative concepts.
>
> We would also like to thank the reviewer again for the insightful comment regarding the overall flow of our claims. We have revised the manuscript to clarify our contributions and marked the changes in blue. Please feel free to reach out if there are any remaining questions or concerns.

---

> ### Author Response · Authors · 2025-08-08
> **Author Response (2/2)**
>
> **Failure Case Analysis and Potential Drawbacks (W2, RC3)**
> > “**(W2)** Adding a uniform extra margin for all hard-negative classes ignores that some pairs may already be well-separated, potentially over-penalising and harming calibration. No analysis of failure cases or calibration metrics is given.
> > ⠀
> > **(RC3)** Discuss potential drawbacks of dynamic mining: Batch-level hard-negative selection could introduce noise or instability. Include a discussion (or experiment) on whether this affects training dynamics, especially under small batch sizes or label imbalance.”
>
> This is a great suggestion. We acknowledge that batch-level hard-negative mining can lead to training instability, particularly under small batch sizes or label imbalance. This issue, however, is not specific to our method but has been widely observed and reported in prior work on hard-negative mining [8, 9]. As our approach also relies on mining hard negatives within mini-batches, it inevitably inherits this limitation. In our experiments, we similarly observed signs of training instability when using small batch sizes (e.g., 128 for miniImageNet), which highlights a broader challenge inherent to these hard-negative-based methods.
>
> Despite this, one of the key advantages of our method lies in its computational efficiency. Although computational cost could also be a potential concern, we observe that our proposed method incurs only a negligible overhead compared to the baseline. As shown in the table below, we measured the epoch runtime by executing both methods concurrently on the same device and averaging the results over five independent trials. This minimal cost arises because our method operates on class prototypes, adding only a small amount of  computations during training. We have added the discussion section including this result in our manuscript.
> | Method   | CIFAR100     | miniImageNet    |
> |----------|--------------|-----------------|
> | Baseline | 46.02s±3.26  | 107.16s±4.27    |
> | Ours     | 46.12s±3.60  | 107.19s±4.18    |
>
> **References**
> > [5] Robinson, Joshua, et al. "CONTRASTIVE LEARNING WITH HARD NEGATIVE SAMPLES." International Conference on Learning Representations (ICLR). 2021.
> > [6] Vasudeva, Bhavya, et al. "Loop: Looking for optimal hard negative embeddings for deep metric learning." Proceedings of the IEEE/CVF International Conference on Computer Vision. 2021.
> > [7] Yue, Yun, et al. "Understanding hyperbolic metric learning through hard negative sampling." Proceedings of the IEEE/CVF Winter Conference on Applications of Computer Vision. 2024.
> > [8] Yuan, Zhuoning, et al. "Provable stochastic optimization for global contrastive learning: Small batch does not harm performance." International Conference on Machine Learning. PMLR, 2022.
> > [9] Balmaseda, Vicente, et al. "Discovering Global False Negatives On the Fly for Self-supervised Contrastive Learning." arXiv preprint arXiv:2502.20612 (2025).

---

### Review · Reviewer_aYjG · 2025-08-01

**Summary Of Contributions:**

The paper addresses catastrophic forgetting and overfitting in few-shot class-incremental learning (FSCIL) by proposing Hard-Negative Prototype Regularization (HNPR). The authors identify that hard-negative samples and classes significantly impact FSCIL performance through empirical analysis, then develop a regularization approach that imposes penalty margins between samples and their most similar class prototypes. Two mining strategies are explored: dynamic selection using model decision boundaries and static selection using pre-trained encoders. Extensive experiments on miniImageNet, CIFAR100, and CUB200 demonstrate state-of-the-art performance with improved intra-class cohesion and inter-class separability.

**Audience:**

Yes

**Claims And Evidence:**

Yes

**Requested Changes:**

Please refer to the Weakness.

**Strengths And Weaknesses:**

### Strengths

- **Compelling empirical motivation**: Provides strong empirical analysis showing hard samples cause disproportionate forgetting in FSCIL (Figure 2), with clear delta analysis demonstrating confusion concentration among a few classes.

- **Comprehensive experimental validation**: Achieves consistent state-of-the-art results across three standard benchmarks with thorough ablation studies, meaningful comparison between mining strategies, and good qualitative analysis.

- **Clear mechanistic insights**: Offers insights through t-SNE visualizations and distance measurements demonstrating improved embedding structure, with effective presentation and reproducible methodology.

- **Practical effectiveness**: Explores both dynamic and static hard-negative selection strategies with thorough comparison, showing consistent improvements across different experimental settings.

### Weaknesses

My main concerns lie in three points:

- **Limited technical novelty**: The core contribution applies well-established hard-negative mining to FSCIL through straightforward modification of angular margin loss. The "dynamic selection" does not truly mine hard negatives since changing prototypes makes selection somewhat arbitrary, differing significantly from established hard-negative mining approaches.

- **Weak theoretical foundation**: No theoretical analysis explains why this specific regularization works or why class-level hard-negative selection outperforms sample-level approaches from metric learning. The claim that dynamic selection introduces beneficial "noise" lacks proper validation and seems speculative.

- **Experimental limitations**: Missing statistical significance testing, computational overhead analysis, and comparison with recent hard-negative mining advances. The static selection underperformance using pre-trained models suggests the approach may not capture meaningful semantic similarities as claimed.

---

> ### Author Response · Authors · 2025-08-08
> **Author Response (1/2)**
>
> We appreciate Reviewer aYjG (R3) for the thorough and valuable comment. We will further clarify our novelty (W1), theoretical foundation (W2), and supplemental experiments (W3) in the following sections.
>
> **Novelty Clarification (W1)**
> > “**(W1)** Limited technical novelty: The core contribution applies well-established hard-negative mining to FSCIL through straightforward modification of angular margin loss. The "dynamic selection" does not truly mine hard negatives since changing prototypes makes selection somewhat arbitrary, differing significantly from established hard-negative mining approaches.”
>
> Thank you for the feedback. We acknowledge that our dynamic selection is different from established hard-negative mining approaches, as we define hard-negatives based on the distance (precisely, cosine similarity) between a sample and prototypes. However, by leveraging class prototypes—a core mechanism in FSCIL—as stable, global proxies rather than transient batch samples, we ground our mining in the model’s holistic view of each class. Updating these prototype similarities at every optimization step allows us to capture “in-the-moment” confusion dynamics, delivering principled regularization without the randomness or noise inherent to sample-level mining. Finally, this prototype-based mining directly mirrors the class-level proxy mining principles of Proxy-NCA [1] or Proxy-Anchor[2], providing a solid theoretical foundation that will be elaborated on in the next section.
>
> **Theoretical Foundation (W2)**
> > “**(W2)** Weak theoretical foundation: No theoretical analysis explains why this specific regularization works or why class-level hard-negative selection outperforms sample-level approaches from metric learning. The claim that dynamic selection introduces beneficial "noise" lacks proper validation and seems speculative.”
>
> This is a great suggestion. Our class-level prototype-based hard-negative selection can be interpreted as proxy-based approaches such as Proxy-NCA and Proxy-Anchor. In Proxy-NCA, each class proxy acts as a representative embedding, and samples are trained to pull their own proxy closer while pushing all other proxies away via a softmax over distances. Proxy-Anchor builds on Proxy-NCA by adding a uniform margin between every sample-proxy pair. In contrast, our HNPR framework inherits the philosophy of proxy-based methods but introduces two key refinements: (i) Selective Margin Application and (ii) Dynamic Proxy Mining.
>
> **Selective Margin Application.** Instead of enforcing the same margin against every non-target proxy, we identify the top-k most confusable class proxies (hard-negatives) and apply an additional margin only to those. Such focused regularization reduces unnecessary constraints on easily separable classes and concentrates learning capacity where it matters most.
>
> **Dynamic Proxy Mining.** We recompute prototype similarities at each optimization step, capturing real-time shifts in the decision boundary. This “in-the-moment” mining ensures that the hardest negatives truly reflect the model’s current confusion, rather than relying on a static set of proxies or noisy per-batch samples.
>
> Together, these elements extend the core proxy-based metric learning paradigm into the incremental, few-shot regime of FSCIL—preserving computational efficiency while delivering more targeted, data-driven regularization. We have clarified this theoretical foundation in the manuscript and marked the changes in blue.

---

> ### Author Response · Authors · 2025-08-08
> **Author Response (2/2)**
>
> **Experimental Limitations (W3)**
> > “**(W3)** Experimental limitations: Missing statistical significance testing, computational overhead analysis, and comparison with recent hard-negative mining advances. The static selection underperformance using pre-trained models suggests the approach may not capture meaningful semantic similarities as claimed.”
>
> Thank you for your comment. We agree that static selection is fundamentally dependent on pre-trained models' representation quality. However, this reflects an inherent limitation of any fixed‐prototype scheme tied to a single pre-trained backbone rather than a flaw in our method, which is precisely why we center our contributions on dynamic selection—an approach that remains robust regardless of initial embedding quality by continuously adapting to evolving class prototypes.
>
> Based on the comments, we have added more experimental results in the manuscript including (i) multiple run on miniImageNet benchmark with five different random seeds to validate the effectiveness of our proposed approach, (ii) evaluation on modern model (i.e., FSCIL-ASP using ViT backbone) and dataset (i.e., ImageNet-R), (iii) comparison with sample-wise hard-negative mining method, and (iv) computational overhead analysis.
>
> **References**
> > [1] Movshovitz-Attias, Yair, et al. "No fuss distance metric learning using proxies." Proceedings of the IEEE international conference on computer vision. 2017.
> > [2] Kim, Sungyeon, et al. "Proxy anchor loss for deep metric learning." Proceedings of the IEEE/CVF conference on computer vision and pattern recognition. 2020.

---

### Author Response · Authors · 2025-08-08
**General Response to Reviewers**

We sincerely thank the reviewers for their valuable and constructive feedback. The reviewers acknowledged that our approach is **straightforward** (R1, R2), **well motivated** (R1, R3), **practical** (R2, R3), and that our comprehensive analyses provide **clear mechanistic insights** (R3).

In response to the comments, we have made the following updates to the manuscript (highlighted in blue):
- **Related work (Section 2)**:
  - Added prior work on hard-example mining-based in few-shot learning (R1, R2)
  - Included theoretical foundations relevant to our proposed method (R3)
- **Experimental Results (Section 5, Appendix)**
  - Added ViT-based FSCIL evaluation (R1)
  - Included multi-run evaluation on miniImageNet (R1)
  - Compared with a sample-wise hard-example mining method (R1, R3)
- **Discussion (Section 6)**:
  - Added discussion on limitations and computational considerations (R1, R2, R3)

Once again, we truly appreciate the reviewers’ insights and thoughtful suggestions that helped improve our work.

---

### Decision · Action_Editor_ZccF · 2025-10-10

**Recommendation:** Accept as is

**Additional Comments:**

Overall, the submission is clearly written, experimentally solid, and well motivated. The reviewers appreciated the improvements made in the revision, including better contextualization and limitation discussion.

Some reviewers have mentioned the novelty issue of this paper, although it is not a key factor to evaluate the quality of the paper according to the acceptance criteria.

Here are some suggestions the authors may consider when preparing the final version of the paper:
1. The authors may provide in-depth analysis and explanation of the results, especially when and why the method is most effective.

2. The proposed method is only applied in the base session; could it be applied in the incremental session to better leverage the new few-shot examples?

**Audience:**

Yes

**Audience Explanation:**

The study addresses a relevant and active topic—improving robustness in FSCIL through embedding regularization and the empirical insights may interest researchers working on continual learning.

**Claims And Evidence:**

Yes

**Claims Explanation:**

The paper introduces a hard-negative prototype regularization framework for few-shot class-incremental learning (FSCIL), supported by well-designed experiments on standard benchmarks such as miniImageNet, CIFAR100, and CUB200. The experiments are carefully executed and demonstrate consistent gains across several benchmarks, the empirical evidence mainly confirms the expected benefit of hard-negative mining—a well-established principle in representation learning.